# Single-atom platinum with asymmetric coordination environment on fully conjugated covalent organic framework for efficient electrocatalysis

Ziqi Zhang [1], Zhe Zhang[1], Cailing Chen [2], Rui Wang[1], Minggang Xie[1], Sheng Wan[1], Ruige Zhang[1], Linchuan Cong[1], Haiyan Lu[1] ✉, Yu Han [3], Wei Xing [4] ✉, Zhan Shi [1] ✉ & Shouhua Feng[1]

Two-dimensional (2D) covalent organic frameworks (COFs) and their derivatives have been widely applied as electrocatalysts owing to their unique nanoscale pore configurations, stable periodic structures, abundant coordination sites and high surface area. This work aims to construct a non-thermodynamically stable $Pt-N_2$ coordination active site by electrochemically modifying platinum (Pt) single atoms into a fully conjugated 2D COF as conductive agent-free and pyrolysis-free electrocatalyst for the hydrogen evolution reaction (HER). In addition to maximizing atomic utilization, single-atom catalysts with definite structures can be used to investigate catalytic mechanisms and structure-activity relationships. In this work, in-situ characterizations and theoretical calculations reveal that a nitrogen-rich graphene analogue COF not only exhibits a favorable metal-support effect for Pt, adjusting the binding energy between Pt sites to H* intermediates by forming unique $Pt-N_2$ instead of the typical $Pt-N_4$ coordination environment, but also enhances electron transport ability and structural stability, showing both conductivity and stability in acidic environments.

Hydrogen is one of the most promising energy sources to replace fossil fuels due to its high energy density and clean combustion products[1]. Electrocatalytic water splitting is a promising approach for hydrogen production, which outperforms mainstream steam reforming process in terms of their zero greenhouse gas emission and use of renewable energy. Therefore, it is of great significance to develop efficient, durable and affordable electrocatalysts with low overpotentials towards electrochemical hydrogen evolution reaction (HER) to drive electrochemical water splitting with low power consumption[2].

Considerable effort has been made in recent years to develop non-noble metal materials to reduce the cost[3]. However, inefficiency, dissatisfactory stability and high overpotential hinder their further applications[4]. So far, platinum (Pt)-based catalysts have been regarded as the most efficient electrocatalysts for HER owing to their optimal adsorption energy towards hydrogen intermediates[5]. However, scarce reserves and the high price of Pt are driving scientists and engineers to develop alternative catalysts with high catalytic activity, cost-effectiveness, and sustainability[6–8].

[1]State Key Laboratory of Inorganic Synthesis and Preparative Chemistry, College of Chemistry, Jilin University, Changchun 130012, P. R. China. [2]Advanced Membranes and Porous Materials Center, Physical Sciences and Engineering Division, King Abdullah University of Science and Technology (KAUST), Thuwal 23955-6900, Saudi Arabia. [3]Electron Microscopy Center, South China University of Technology, Guangzhou 510640, P. R. China. [4]State Key Laboratory of Electroanalytical Chemistry, Changchun Institute of Applied Chemistry, Chinese Academy of Sciences, Changchun 130022, P. R. China. ✉e-mail: luhy@jlu.edu.cn; xingwei@ciac.ac.cn; zshi@mail.jlu.edu.cn

As a desirable strategy to maximize atomic utilization, fabrication of single-atom catalysts (SACs) can be a possible solution to the above challenge[9,10]. Furthermore, conductive agent-free and pyrolysis-free SACs with definite structures can be employed to investigate catalytic mechanisms and structure-activity relationships, which is of great significance for the in-depth understanding and subsequent development of electrocatalysts with highly efficient active sites. However, SACs are prone to aggregate owing to their high surface energy, which dilutes the number density of active sites and suppresses activities[11]. Therefore, it is necessary to stabilize single Pt atoms on a desirable substrate which can tune the electronic structure of Pt sites via metal-support interaction[12]. It will improve catalytic activity and provide robust coordination environment to prevent aggregation or shedding of Pt sites[13].

2D materials such as graphene[14], layered double hydroxides (LDHs)[15], and MOFs[16] have been utilized as substrates for loading noble-metal single atoms owing to their large specific surface area, high reactant accessibility, and desirable metal-support interactions. However, the introduction of graphene or other conductive agents leads to the uncertainty of active center or damage the catalytic structure[17]. Worse still, most of LDHs and MOFs are vulnerable in acidic environments, which limits further applications in the field of energy conversion, such as polymer electrolyte membrane (PEM) water electrolyzer[18]. In addition, the metal centers tend to form thermodynamically stable M-$N_4$ or M-OH (M=metal center) in MOFs or LDHs synthesized by the traditional crystallization method, which reduces their tunability and flexibility[19]. Recently, low-coordinated Pt SACs with broken geometric symmetry, instead of Pt−$N_4$ or Pt−$S_4$ stabilized by a near-perfect square planar geometry ($D_{4h}$), have been proved to achieve higher intrinsic activity experimentally and theoretically[20]. Therefore, an uncommon type of Pt-$N_2$ unsaturated coordination active site was first constructed onto an acid-proof 2D substrate by an electrochemical method in this work, aiming to break through the upper limit of intrinsic activity by creating a new coordination configuration.

Recently, covalent organic frameworks (COFs) and their derivatives have been widely reported as electrocatalysts[21,22], including HER[23] and OER[24], owing to their environment-friendly nature, earth-abundant, cost-effective, and resistance to a wide pH range[25]. However, most of COF or COF-derived electrocatalysts did not perform well in electrocatalytic water splitting due to their poor electrical conductivity[26,27]. The two widely used methods of increasing electrical conductivity are carbonization under high temperature[28] and doping conductive agent[29], which may damage the structure of catalysts. If the structural accuracy is lost, it is a great challenge to unveil catalytic mechanisms or determine the structure-activity relationship. Heterogeneous dispersion of conductive agents introduced subsequently through top-down synthetic strategy in COFs is common[30], which will significantly reduce the electron transport capacity at the interfaces between active sites and conductive agents or current collectors[31]. As a result, although COF-derived active centers are theoretically process high catalytic activity and ideal adsorption energy towards reactant intermediates, the electrons among these active sites have no access to the current collection or working electrode, which leads to unsatisfactory catalytic performance.

The fully conjugated 2D COFs could facilitate the efficient electron delocalization and transport within the extended π-conjugated backbones[32], exhibiting synergistic features of semiconducting properties[33] combined with porous graphene-like layered structure[30]. The entire covalent bond linkage enables high chemical stabilities whilst the fully conjugated graphene-like layered structure gives it satisfactory conductivity through high degree of π-conjugation[34]. Therefore, it can serve as an ideal substrate to accommodate Pt atoms owing to its unique nanoscale pore configuration, stable periodic structure, abundant coordination sites, enhanced conductivity, and

high surface area. It imposes a great potential to provide numerous active sites towards HER and desirable long-term stability against harsh pH environments[35,36].

In this work, we fabricate a conductive agent-free and pyrolysis-free Pt-based SAC at room temperature and pressure by electrochemically modifying Pt single atoms into a 2D nitrogen-rich graphene analogue COF (NGA-COF@Pt)[37]. Through a bottom-up method, Pt single atoms are anchored onto the highly π−π conjugated graphene-like COF homogeneously (Pt load ≈ 2.66 wt.%), which not only facilitates charge transfer by using COF itself as a homogeneous current collector but also enhances the intrinsic activity of Pt single atoms by forming optimized metal-support interactions[38]. The synthesized NGA-COF@Pt exhibits satisfactory catalytic performance and atomic utilization in universal pH environments. The overpotentials needed for driving HER at current density of 10 mA cm$^{-2}$ are 13 mV in 0.5 M $H_2SO_4$ and 19 mV in 1 M KOH, ranking it one of the most efficient HER catalysts reported to date[39]. Furthermore, a series of theoretical and experimental studies reveal that Pt-$N_2$ active site is fabricated by the electrochemical modification method rather than the most thermodynamically stable M-$N_4$ active site. Such an unsaturated coordination mode greatly improves the intrinsic activity of every Pt single active site by achieving high mass activity (18165 A $g_{Pt}^{-1}$) and turnover frequency (TOF, 18.4 s$^{-1}$) at overpotential of 44 mV. In all, NGA-COF regulates the adsorption energy of Pt for H* intermediates by forming unique Pt-$N_2$ coordination environment while the fully conjugated 2D structure improves the electron transport capacity of the active sites through the path of Pt-N-C, which supports direct electron transfer from the catalytic active sites to electrode homogeneously. The complementary metal-support interaction between Pt single atoms and NGA-COF significantly improves the conductivity and intrinsic activity of the catalyst without extra conductive agent or pyrolysis. Moreover, by using in-situ deuterium (D) isotope labeling differential electrochemical mass spectrometry (DEMS), we provide direct experimental evidence that Pt particles will be poisoned by HER intermediates while Pt single atoms with asymmetric coordination environment will not.

## Results

### Electrocatalyst synthesis
NGA-COF with high crystallinity was synthesized by optimizing parameters based on NGA-CMP, which was previously reported by our group[37]. Though both have the same in-plane structure and component monomers, the former has higher crystallinity which furnish it with a more orderly stacking pattern along the c-axis. It guarantees structural accuracy in the subsequent single-atom modification process. As schematically illustrated in Fig. 1a, NGA-COF was synthesized by a reaction of 2,3,6,7,10,11-triphenylenehexamine (TPHA) and hexaketocyclohexane (HKH). The synthesized NGA-COF has a regular 2D layered structure with an interlayer distance of 3.37 Å. Transmission electron microscopy (TEM, Fig. 1b) image depicts the graphene-like layered structure of NGA-COF along c-axis. XRD pattern was collected to identify the crystallographic characteristics of NGA-COF. The sharp diffraction peaks at 2θ = 7.2 °C and 26.4 °C (Fig. 1c) are assigned to the (100) and (001) facets of NGA-COF, respectively, corresponding to the periodic pore structure in the plane and the distance of 3.37 Å (d-spacing) between the conjugated (001) planes of NGA-COF. The absence of the characteristic (101) peak in the experimental XRD pattern confirms the AA-stacking structure of the synthesized NGA-COF, thereby ruling out the possibility of AB-stacking. FTIR spectrum of NGA-COF (Supplementary Fig. 1a) proves the formation C−N and C=N bonds as the typical stretching vibration peaks at 1300 and 1628 cm$^{-1}$. Besides, four signal peaks appeared in the $^{13}$C cross-polarization magic angle spinning (CP/MAS) NMR spectra (Supplementary Fig. 1b), corresponding to the four chemical environments of the carbon element of NGA-COF, which is consistent with literature[30]. $N_2$ adsorption−desorption was carried out to

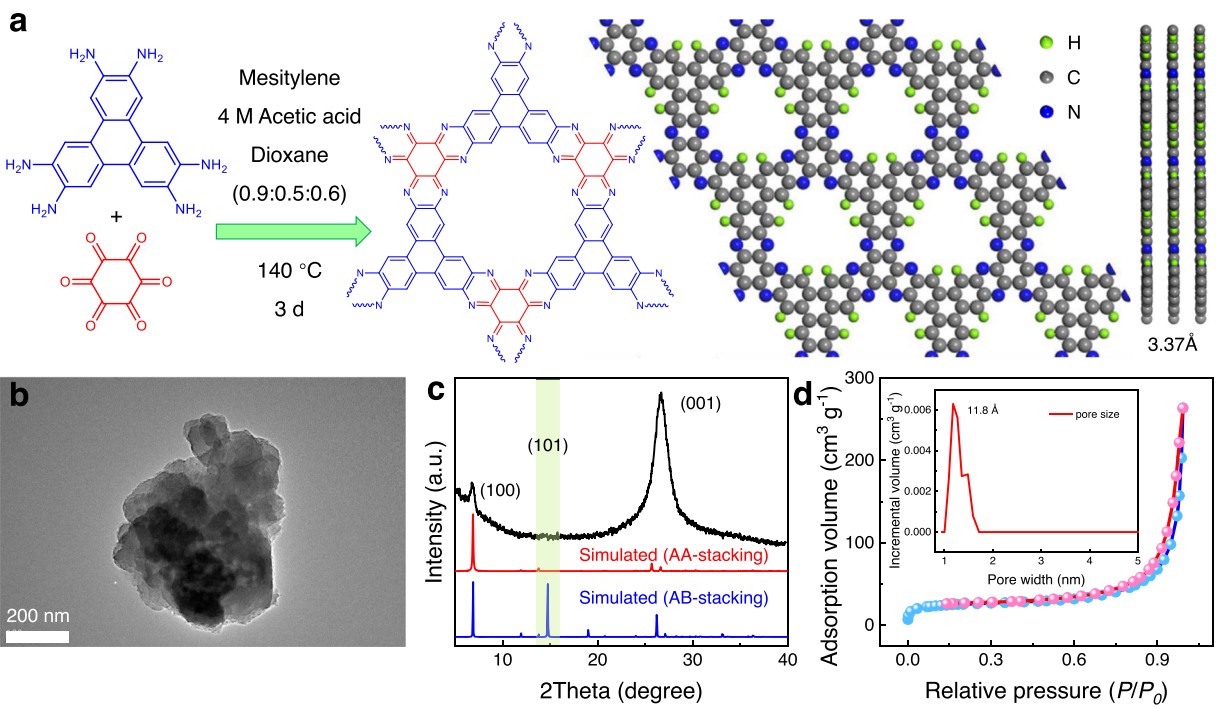

**Fig. 1 | COF synthesis and physical characterization. a** Schematic illustration of the synthesis of NGA-COF. **b** TEM image of NGA-COF. **c** XRD patterns of NGA-COF synthesized and simulated. **d** Nitrogen adsorption−desorption isotherm of NGA-COF with corresponding pore size distribution inset.

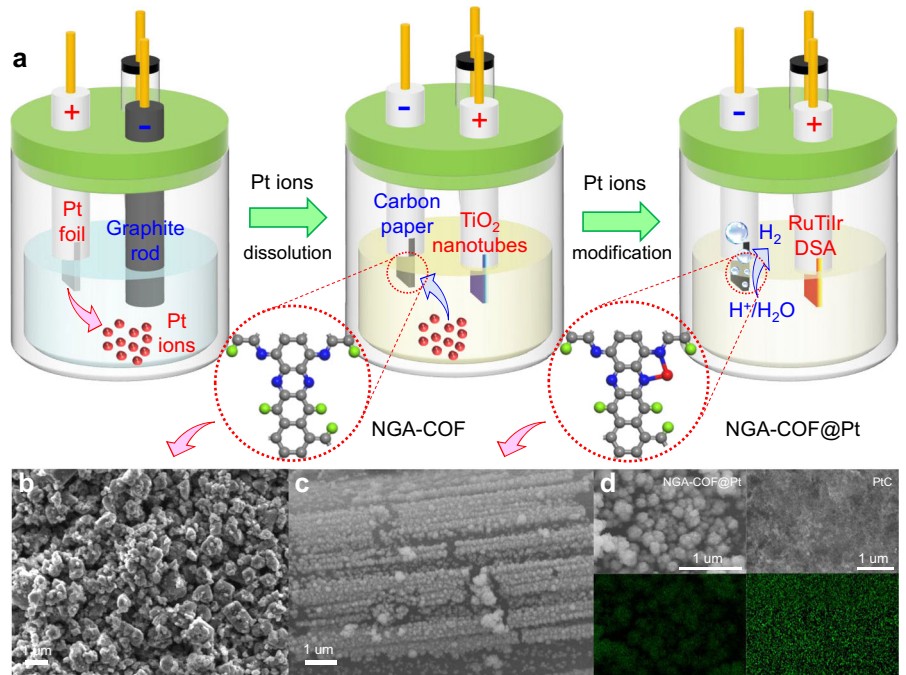

**Fig. 2 | Electrocatalyst synthesis and morphology characterization. a** Schematic illustration for preparation of NGA-COF@Pt catalyst in-situ on carbon paper electrode by electrochemical modification. **b** SEM image of NGA-COF. **c** SEM image of NGA-COF@Pt arrays deposited on carbon paper. **d** Corresponding EDS mappings of NGA-COF@Pt and commercial 20 wt.% PtC for Pt elements.

quantify the specific surface area and pore structure of NGA-COF. Results revealed the dominant micropores with a pore width of 11.8 Å in NGA-COF with the calculated Brunauer−Emmett−Teller (BET) surface area about 86.3 m$^2$ g$^{-1}$ (Fig. 1d), which agrees with the pore size of the simulated COF-NGA structure (Supplementary Fig. 2).

Next, the mild electrochemical modification process was carried out in 0.5 M H$_2$SO$_4$ solution (Fig. 2a) through a certain number of cyclic

voltammetry (CV) segments to anchor Pt single atoms into NGA-COF to fabricate NGA-COF@Pt with the aid of a sheet of titanium with TiO$_2$ nanotubes on surface (TiO$_2$ NTs, see supporting information and Supplementary Fig. 3) as counter electrode (CE). The synthesized NGA-COF@Pt on carbon paper substrate was further applied as a cathode for electrochemical water splitting (EWS). Scanning electron micro-scopy (SEM, Fig. 2b) demonstrates that the uniform particle size of

NGA-COF is less than 1.0 um. Some lumpy or layered structures with larger particle sizes appear mainly owing to the polymerization of small COF particles. High-resolution transmission electron microscopy (HRTEM, Supplementary Figs. 4 and 5) images demonstrated the layered graphene-like structure of NGA-COF before electrochemical modification. After electrochemical deposition, NGA-COF@Pt catalyst was uniformly deposited on carbon paper substrate in the form of regular microcosmic arrays with the aid of the oriented electric field force from the TiO$_2$ NTs CE[40] (Fig. 2c). Energy-dispersive spectroscopy (EDS) mapping (Fig. 2d) reveals uniform distribution of Pt element among the nanoparticles, suggesting that the Pt atoms were anchored among NGA-COF. In addition, inductively coupled plasma–optic emission spectrometry (ICP-OES) confirms that 2.66 wt.% Pt was loaded in NGA-COF@Pt.

## Electrocatalyst characterization

To verify the existence of Pt single atoms in NGA-COF@Pt and reveal the coordination environment precisely, aberration-corrected high-angle annular dark-field scanning transmission electron microscopy (HAADF-STEM) and X-ray absorption fine structure (XAFS) were applied to determine the element bonding configuration of NGA-COF@Pt[41]. The presence of abundant Pt single atoms is confirmed by aberration-corrected HAADF-STEM (Fig. 3a). Brighter spots assigned to Pt presented atomically dispersion among NGA-COF substrate (Supplementary Fig. 6).

Based on normalized Pt L$_3$-edge X-ray absorption near-edge structure (XANES) spectra (Fig. 3b), the white-line peak for NGA-COF@Pt is higher than that of Pt-foil while lower than that of PtO$_2$,

indicating that the valence state of Pt in NGA-COF is between Pt$^0$ and Pt$^{4+}$[42]. Different from Pt-foil or PtO$_2$ in which Pt-Pt or Pt-O peak is dominant, only the peak at ca. 1.7 Å (all peaks above were not corrected with phase shift) appears in NGA-COF@Pt (Fig. 3c), suggesting Pt single atoms are coordinated with N in NGA-COFs[43]. Most importantly, no Pt-Pt signal exists in FT EXAFS, which indicates that single Pt atoms are present in NGA-COF@Pt.

To confirm the coordination environment of Pt in NGA-COF@Pt, quantitative EXAFS curve fittings were performed between synthesized and simulated NGA-COF@Pt (Fig. 3d, e and Supplementary Table 1)[44]. The fitting result demonstrates that the Pt single atom was anchored on NGA-COF@Pt by coordinating with two adjacent nitrogen atoms (Pt-N$_2$)[45]. The subsequent experiments and calculations demonstrate that the unique metal-support interaction between Pt and NGA-COF greatly increases the catalytic activity and electron transport efficiency[46].

To reveal more atomic structure information of NGA-COF@Pt, wavelet-transform (WT) analysis was carried out to provide powerful resolution in both k and R spaces (Fig. 3f–h)[47,48]. The maximum intensity of WT at 5.53 Å$^{-1}$ (k space) is adequately resolved at 1.51 Å (R space) in NGA-COF@Pt, which is ascribed to the Pt-N$_2$ coordination structure. The maximum intensity at 11.57 Å$^{-1}$ and 7.41 Å$^{-1}$ corresponds to Pt–Pt and the absence of Pt-O coordination for the reference samples of Pt-foil and PtO$_2$ further confirms the loading of Pt single atoms into NGA-COF conductive network.

With prolonged electrochemical modification, Pt particles appear (Supplementary Fig. 7) and the lattice diffraction stripes of Pt (111) are apparent by HRTEM when the CV segments increase to 2500 from

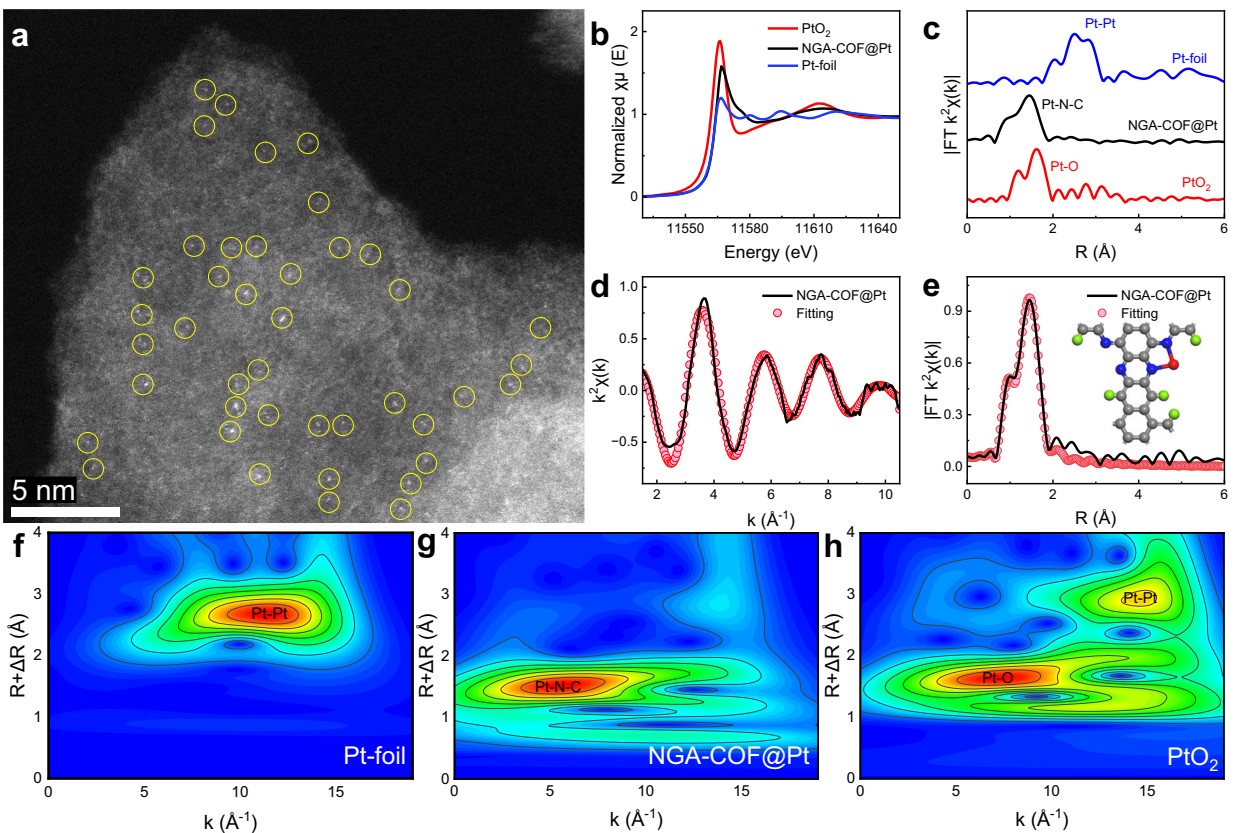

**Fig. 3 | Coordination environment of electrocatalysts. a** Atomic-resolution aberration-corrected HAADF-STEM image of NGA-COF@Pt showing the atomically dispersed Pt atoms, which are highlighted by yellow solid line circle. **b** Pt L$_3$-edge XANES spectra and **c** k$^2$-weighted FT-EXAFS spectra of Pt-foil, NGA-COF@Pt and PtO$_2$. Corresponding EXAFS **d** k and **e** R space fitting curve of NGA-COF@Pt (Inset: Simulations models of NGA-COF@Pt). WT-EXAFS signals of **f** Pt-foil, **g** NGA-COF@Pt and **h** PtO$_2$.

2000 (Supplementary Fig. 8). Theoretically, NGA-COF@Pt-NP (ICP-OES confirms that 4.32 wt.% Pt was loaded in NGA-COF@Pt-NP) is thermodynamically more stable than NGA-COF@Pt (Supplementary Fig. 9), indicating that the increasing CV segments transforms the latter to the former.

Therefore, electrochemical modification was applied to anchor single Pt atoms into NGA-COF, since this method can control the reaction process by adjusting specific and quantifiable parameters such as voltage, current, and deposition time[49]. In conclusion, a large number of CV segments cause Pt single atoms to aggregate to particle form, resulting in the decreasing active sites and degradation of catalytic performance (Supplementary Fig. 10).

Density functional theory (DFT) calculations and X-ray photoelectron spectroscopy (XPS) reveal electronic structure between NGA-COF and a single Pt atom chemically bonded. Electron localization function (ELF) and Bader charge analysis confirm that a single Pt atom will attract electron clouds from the two coordinated N atoms (Fig. 4a, b). Similarly, charge density differences calculations demonstrate that the electrons of N are attracted to Pt site (Fig. 4c), forming a Pt←N electron pathway[50]. High-resolution XPS spectra of C 1s (Fig. 4d) depict that the chemical environments of C remain after the introduction of Pt, indicating that Pt was not directly coordinated with the C sites. The appearance of a carbon peak at the binding energy of about 292.6 eV is mainly ascribed to the carbon paper substrate rather than the introduction of Pt (Supplementary Fig. 11a). On the other hand, the binding energy of N significantly increases after the introduction of Pt (Fig. 4e), suggesting that the introduced Pt atoms are coordinated with N rather than O (Supplementary Fig. 11b)[51], which is consistent with the previous XAS analysis. In addition, the increasing binding energy of N element indicates the shifts of electron clouds of N toward Pt, agreeing with the DFT charge analysis (Fig. 4a–c). The phenomena mentioned above can not only increase the activity of Pt sites through desirable metal-support interactions, but also enhance the adsorption ability towards the catalytic precursor $H^+$ and $H_3O^+$ by boosting the electronegativity of Pt[52]. In addition, the high-resolution spectra of Pt 4f depict that, compared to the coordination with C in PtC, Pt atoms coordinated with N in NGA-COF@Pt exhibited lower binding energy (Fig. 4f), indicating a larger electron cloud density and electronegativity. In conclusion, this Pt electronic structure optimized and adjusted by NGA-COF support exhibits high HER performance both experimentally and theoretically.

## Electrocatalytic hydrogen evolution reaction performance

HER catalytic performance was tested in 0.5 M $H_2SO_4$ solution. The synthesized NGA-COF@Pt presents a lower overpotential (13 mV) at 10 mA cm$^{-2}$ than noble-metal benchmark commercial 20 wt.% PtC (Fig. 5a and Supplementary Fig. 12). As a single-atom catalyst, NGA-COF@Pt exhibits high mass activity (18165 A g$_{Pt}^{-1}$) and TOF (18.4 s$^{-1}$) at an overpotential of 44 mV. These performances surpass commercial PtC (1013 A g$^{-1}$ and 1.02 s$^{-1}$) and some other noble-metal-based HER catalysts under the same condition (Fig. 5b and Supplementary Fig. 13). It is worth noting that due to different mass loadings and mass transport limitations, it is difficult to directly compare the performance of prepared catalysts with each other and to PtC[53]. To get further insights into the kinetics and mechanism for HER, Tafel plots based on the corresponding polarization curves are presented in Fig. 5c. The lower Tafel slope of NGA-COF@Pt (21.88 mV dec$^{-1}$) compared to PtC (28.65 mV dec$^{-1}$) suggests higher HER kinetics. Meanwhile, electrochemical impedance spectra (EIS) were employed to understand the charge transfer resistance of NGA-COF, NGA-COF@Pt, and PtC. The electrochemical modification process reduces the charge transfer resistance ($R_{ct}$) of COF from 60.5 Ω to 20.1 Ω by introducing Pt atoms into NGA-COF (Fig. 5d and Supplementary Table 2). Furthermore, though PtC has the smallest electron transfer resistance of all the samples, NGA-COF@Pt has higher catalytic activity mainly because of a higher intrinsic activity. It was proved by subsequent electrochemically active surface area (ECSA) tests (Supplementary Fig. 14) and theoretical calculations. Due to the uniformly smooth two-dimensional planar structure, NGA-COF@Pt exhibits a smaller electrochemical active surface area than a rough PtC surface. However, through the exfoliation process caused by ultrasonic and electrochemical modification (Supplementary Fig. 7)[6], the 2D nanosheets derived from COF can greatly improve the utilization of Pt noble metal by completely exposing the active sites with an extremely high ECSA normalized current density. Chronopotentiometric tests reveal that the stability of the synthesized NGA-COF@Pt exceeds that of commercial PtC by a smaller voltage rise (Supplementary Fig. 15). SEM and TEM images of NGA-COF@Pt demonstrate that the characteristic microscopic array

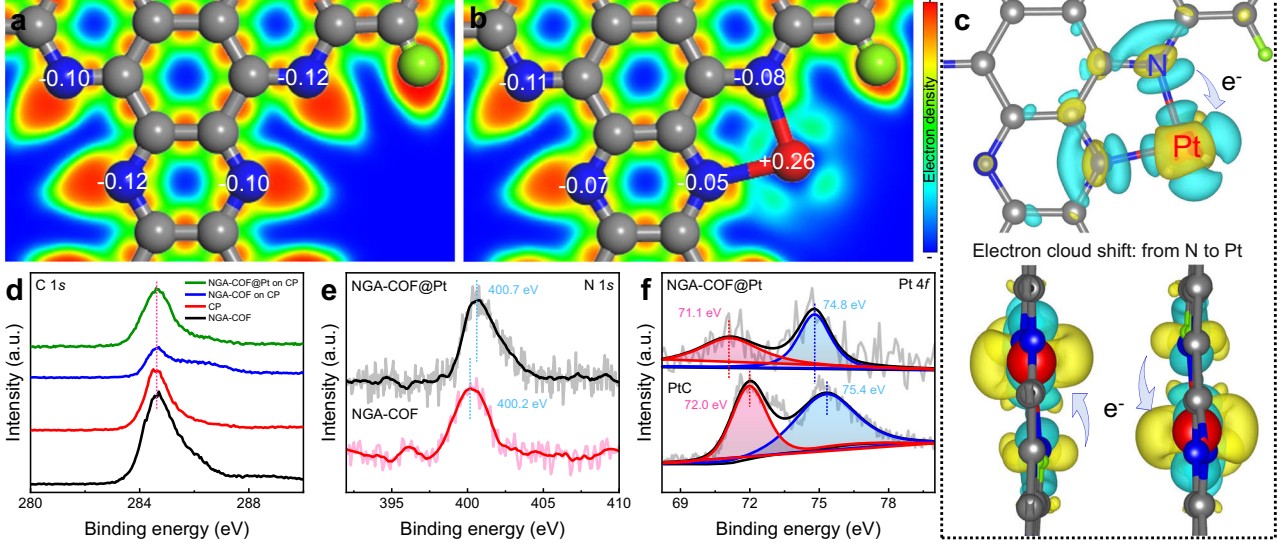

**Fig. 4 | Geometric and local electronic structures of electrocatalysts.** ELFs of **a** NGA-COF and **b** NGA-COF@Pt with Bader charge analysis marked on specific atoms. The H, C, N, and Pt elements are shown in green, grey, blue and red, respectively. **c** The differential charge density distribution map of NGA-COF@Pt along the **c** (upper), a (bottom left) and **b** (bottom right) axis, where the isosurface value is set to be 0.005 e Å$^{-3}$ and the positive and negative charges are shown in yellow and cyan, respectively. High-resolution XPS spectra of **d** C 1s, **e** N 1s, **f** Pt 4f of different samples.

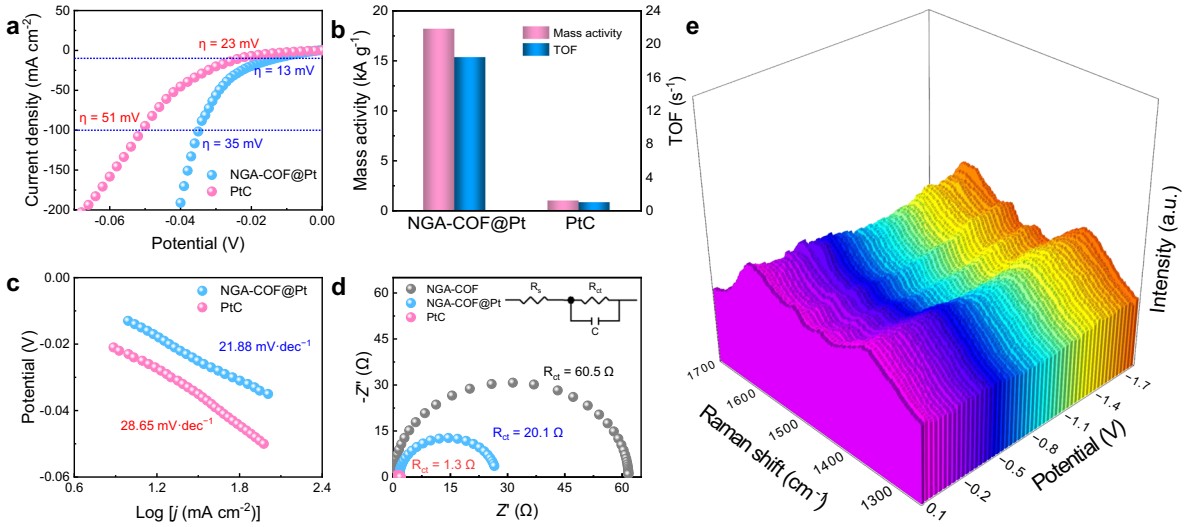

**Fig. 5 | Electrocatalytic hydrogen evolution reaction performance. a** Polarization curves, **b** mass activity and TOF, **c** corresponding Tafel plots and **d** EIS of different samples (inset shows the equivalent circuit diagram). **e** In-situ Raman spectra of NGA-COF@Pt at various potentials. Electrolyte: 0.5 M $H_2SO_4$.

morphology of NGA-COF@Pt remains after the stability test (Supplementary Fig. 16) under the high current density.

In-situ Raman spectro-electrochemistry was employed to monitor the structure transformation of NGA-COF@Pt during the HER process (Fig. 5e and Supplementary Fig. 17). Typically, Raman shift at around 1356 cm$^{-1}$, 1462 cm$^{-1}$, 1513 cm$^{-1}$, and 1610 cm$^{-1}$ can be assigned to the characteristic peaks of the benzene ring, channel relaxation, C = C and C = N vibration, respectively[54]. Notably, the intensity of the benzene ring and channel relaxation peaks did not change during the HER process, indicating that the graphene-like layered structures remained. The intensity of the characteristic vibration peak of C = N remains even at a low potential (−1.7 V), indicating that the coordination mode of Pt-N$_2$ is intact. In general, the structure of NGA-COF@Pt remained during the HER process owing to its unique rigidity, aromaticity, and strong π-π interaction of NGA-COF.

**Catalytic mechanisms and structure-activity relationships**

Apart from the acidic environment (0.5 M $H_2SO_4$), the HER catalytic performance was also tested in an alkaline (1 M KOH, Supplementary Figs. 18–24) and in a neutral environment (0.1 M PBS, pH=7.4, Supplementary Fig. 25). In all, NGA-COF@Pt exhibited outstanding HER performance under alkaline conditions and under neutral conditions, outperforming commercial benchmark PtC (Fig. 6a).

To uncover the mechanism for HER, DFT was applied to calculate the free energy of H* intermediates, which has been proven to be key evidence for characterizing the HER activity of electrocatalyst[55]. NGA-COF has lower absolute adsorption energy towards H* than pristine NGA-COF@Pt (Fig. 6b and Supplementary Fig. 26). Further calculations show that NGA-COF@Pt is prone to bond with the first two H atoms ($\Delta G_{H*} = -1.217$ eV and $-1.254$ eV) to form NGA-COF@Pt-2H (Supplementary Fig. 27) and then NGA-COF@Pt-2H exhibits desirable intermediate adsorption energy ($\Delta G_{H*} = -0.019$ eV) close to 0 eV. Some other possible coordination modes and active sites were also calculated by DFT (Fig. 6c, Supplementary Figs. 28–31 and Supplementary Table 3), but the calculated adsorption energy towards H* is far from the experimental value (Atomic coordinates of the optimized computational models have been provided in the Supplementary Data 1). Therefore, the most likely active site was Pt atom with asymmetric coordination environment (Pt-N$_2$H$_2$) of NGA-COF@Pt-2H because its theoretical overpotential is closest to the overpotential measured in

the electrochemical experiments mentioned above. The desirable adsorption energy of NGA-COF@Pt-2H towards the H* intermediate will greatly improve the mass transfer efficiency during the reaction and thus increase TOF to 18.4 s$^{-1}$. This value is well-placed among current mainstream Pt single-atom catalysts (Supplementary Tables 4 and 5). This is also the reason why the Tafel slope of NGA-COF@Pt (21.88 mV dec$^{-1}$, Fig. 5c) is lower than the theoretical value of Volmer-Tafel reaction mechanism (28.5 mV dec$^{-1}$)[56].

Experiments were conducted to prove that catalysis happens at Pt of NGA-COF@Pt-2H rather than NGA-COF@Pt. First, at the Pt L$_3$ edge, the Pt white line spectrum of NGA-COF@Pt broadened to higher energy (Supplementary Fig. 32), indicating that H atoms were bonded to Pt atoms[57]. Second, Zeta potential measurements conducted at different pH values (Fig. 6d) confirm that NGA-COF is readily protonated at low pH conditions given its high zeta potential[14]. However, after Pt coordination, the zeta potential of NGA-COF@Pt significantly decreases compared with that of NGA-COF. It is ascribed to the stabilization of electrons on N atoms through the coordination between Pt and N, which thus makes it harder to be protonated. In addition, according to the theoretical calculation results (Supplementary Fig. 33), during the zeta potential test, if Pt is only coordinated with two N, then it must be coordinated with 2 Cl atoms in KCl solution to form the structure of NGA-COF@Pt-Cl2. However, the signal of Cl was invisible after Zeta potential measurements (Supplementary Fig. 34), indicating that before the Zeta potential test, the unoccupied coordination orbital of Pt had been saturated with two H, forming NGA-COF@Pt-2H. Third, a comparison of in-situ Raman spectra in acidic (Fig. 5e) and neutral (Supplementary Fig. 35) conditions reveals that the C = N vibration peak at 1610 cm$^{-1}$ remained unchanged with the decease of the applied voltage since the coordination of Pt rendered the bonded N unable to be protonated. It indicates that the formation of Pt-H occurs after the electrochemical modification and before the HER test. Based on the above arguments, it can be concluded that the actual configuration with catalytic activity obtained just after electrochemical modification is NGA-COF@Pt-2H. NGA-COF@Pt-2H is directly employed as a catalyst during the HER process (Supplementary Fig. 36).

To understand the origin of catalytic activity and the improvement of electron transport ability after the combination of Pt single atoms and NGA-COF from a theoretical perspective, total density of

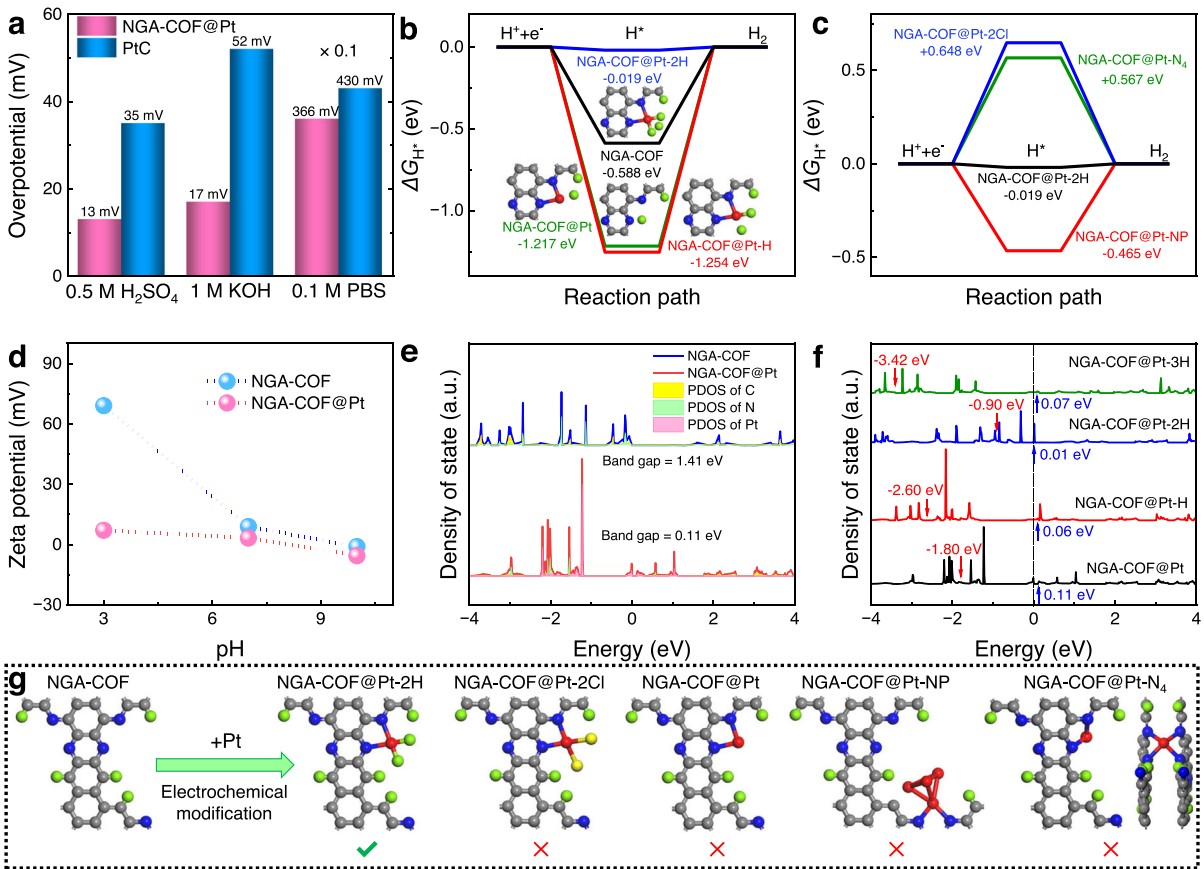

**Fig. 6 | Configuration of the active site and catalytic mechanism determined by DFT calculations. a** Overpotentials needed to drive HER at the current density of 10 mA cm$^2$ in different electrolytes for NGA-COF@Pt and PtC. **b** Free energy diagram of HER over N site of NGA-COF and Pt sites of NGA-COF@Pt, NGA-COF@Pt-H, and NGA-COF@Pt-2H. **c** Free energy diagram of HER over Pt sites of NGA-COF@Pt-2Cl, NGA-COF@Pt-N$_4$, NGA-COF@Pt-2H, and NGA-COF@Pt-NP. **d** Zeta potential of NGA-COF and NGA-COF@Pt at different pH values. **e** Calculated TDOS of NGA-COF and NGA-COF@Pt. **f** Calculated TDOS of NGA-COF@Pt, NGA-COF@Pt-H, NGA-COF@Pt-2H, and NGA-COF@Pt-3H marked with d band centers (red arrows) and band gaps (blue arrows). **g** Schematic illustration of the possible coordination environment for Pt element after being electrochemical modified onto NGA-COF.

states (TDOS) and partial density of states (PDOS) were calculated to estimate the energy band transformation and d-band center variation around the Fermi level. First, NGA-COF exhibits a wide range of forbidden bands (1.41 eV) around the Fermi level so no active electrons participate in the catalysis of HER (Fig. 6e). Fortunately, after the doping of Pt single atoms, the band gap was reduced to 0.11 eV and the connectivity of energy band was improved significantly (Supplementary Fig. 37). The addition of Pt optimizes the overall electron connectivity near the active site since PDOS of C and N elements also contribute to the electronic states near the Fermi level (Fig. 6e). In other words, NGA-COF regulates the adsorption energy of Pt for H* intermediates (Fig. 6b), while Pt single atom improves electron transport capacity of the local Pt-N-C. And these Pt-N-C localities, as catalytic active units, are directly connected to the entire NGA-COF conductive network so that the electrons can be transferred directly from the catalytic active sites to the electrode homogeneously without passing through a foreign conductive agent with a heterogeneous structure. Such complementary metal-support interaction between Pt single atoms and NGA-COF greatly improves the conductivity of materials (Fig. 5d and Supplementary Fig. 18d) without the aid of extra conductive agent or py-rolysis. TDOS of NGA-COF@Pt with different numbers of H atoms attached is calculated (Fig. 6f), indicating that NGA-COF@Pt-2H has the smallest band gap (0.01 eV) and the highest d-band center (−0.90 eV) among all the samples. Theoretically, the smallest band gap leads to the highest conductivity while the highest

d-band center reflects the highest energy of anti-bonding orbitals formed by Pt 5d and H 1s, which enhances the adsorption and transformation of reactant intermediates so as to improve the catalytic activity of Pt sites in NGA-COF@Pt-2H[58]. In addition, the band gap of NGA-COF@Pt-NP is wider than that of NGA-COF@Pt and NGA-COF@Pt-2H (Supplementary Fig. 38), which also explains that the formation of particles (Supplementary Figs. 7–9) decreases catalytic performance (Supplementary Fig. 10) from the perspectives of electronic state. NGA-COF@Pt-2H has a smaller energy matching degree for the adsorbed H* intermediates than those with NGA-COF@Pt-H and NGA-COF@Pt (Supplementary Fig. 39), which facilitates the desorption process of the generated H$_2$ from Pt sites of NGA-COF@Pt-2H. In summary, it is demonstrated from a theoretical point of view that NGA-COF@Pt-2H is the most desirable and realistic catalytic configuration of all (Fig. 6g).

According to the previous research[59], H$_2$ is prone to underpotential deposits on the surface of Pt (111). The Gibbs free energy ($\Delta G_0$) of the process where H$_2$ molecules adsorbed to Pt is positive, which is significantly higher than that of the dissociation process. In other words, it is thermodynamically difficult for H$_2$ to bond with Pt in its complete molecular form. Instead, H$_2$ molecules will dissociate into H atoms and spontaneously adsorb on the surface of Pt (111), a thermodynamically spontaneous process that has been theoretically verified and more recently observed and confirmed experimentally[60]. In short, the H atoms will adsorb to Pt when they initially impact defect

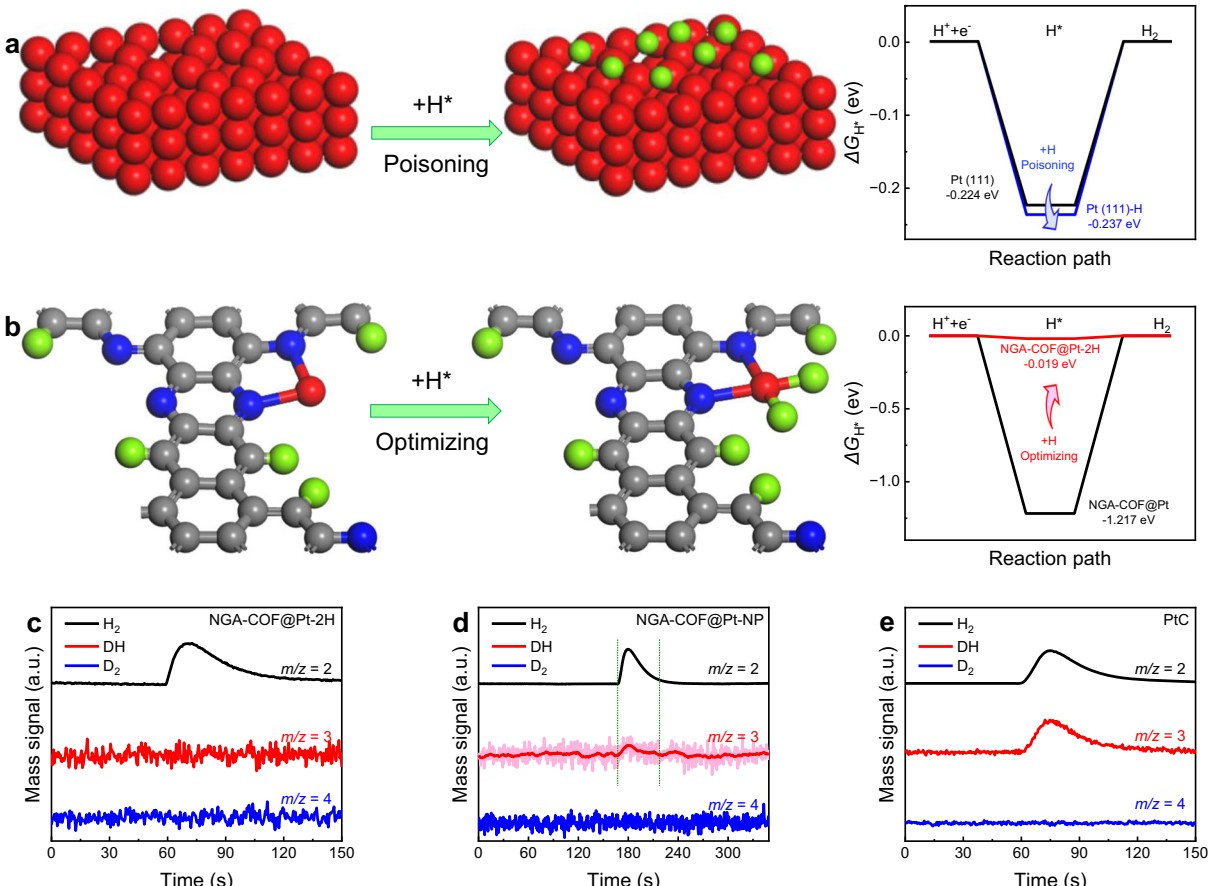

**Fig. 7 | Poisoning mechanism analysis.** Schematic illustration and corresponding free energy diagram of HER over Pt site of **a** Pt (111) and **b** NGA-COF@Pt before and after optimizing by H* intermediates. In-situ DEMS measurements of $H_2$, DH and $D_2$ signals from the HER products for D-labeled **c** NGA-COF@Pt-2H, **d** NGA-COF@Pt-NP and **e** PtC in 0.5 M $H_2SO_4$ in $H_2O$.

sites among the Pt (111) crystal plane, the tendency of which is strong and spontaneous both theoretically and experimentally. These findings have inspired us to consider that if a HER catalyst contains the Pt (111) crystal plane, such as NGA-COF@Pt-NP synthesized in this work (Supplementary Fig. 7) and commercial PtC, it may be poisoned by HER reaction intermediates because this specific adsorption of H atoms towards Pt (111) is inevitable and the desorption is difficult. To verify this conjecture theoretically, DFT calculations were performed on the adsorption-free energy of HER intermediates over different Pt sites before and after hydrogenation. For Pt (111) (Fig. 7a), binding energy of Pt site towards H* deteriorate from −0.224 eV to −0.237 eV after hydrogenation [Pt (111) poisoned by H atoms is referred to as Pt (111)-H]. In contrast, for NGA-COF@Pt (Fig. 7b), the binding energy of Pt site towards H* advance from −1.217 eV to −0.019 eV after endowing Pt-$N_2$ configuration with two H atoms. In summary, DFT calculations theoretically confirm that hydrogenation poisons the traditional Pt (111) site while optimizes the Pt-N2 site of the NGA-COF@Pt electrocatalyst. In addition, NGA-COF@Pt-2H, of which Pt forms a near-perfect square planar geometry with two coordinated H atoms, not only prevents the generation of Pt (111) crystal surface and its corresponding defects but also avoids the over-strong specific adsorption of H* intermediate through the H-saturated Pt tetra coordination crystal field ($D_{4h}$). In other words, H atoms that may be adsorbed at the Pt sites of NGA-COF@Pt-2H during HER will be more easily detached because Pt tends to recover from unstable Pt-$N_2H_3$ to Pt-$N_2H_2$ tetrahedral configuration.

To verify that the Pt particles will be poisoned by HER intermediates while NGA-COF@Pt-2H will not, in-situ D isotope-labeling DEMS measurements were performed. First, to eliminate the influence of the natural abundance of D and D potentially bounded with CP substrate, in-situ DEMS was first carried out on bare CP (Supplementary Fig. 40a). Only $m/z = 2$ signal ($H_2$) was detected during HER, suggesting that the natural abundance of D and CP substrate will cause no error in subsequent tests. Second, NGA-COF was tested, and no signal of DH or $D_2$ was collected (Supplementary Fig. 40b), ruling out the possibility of NGA-COF being protonated by D (Fig. 6d). Next, in-situ DEMS was conducted on NGA-COF@Pt-2H (Fig. 7c), NGA-COF@Pt-NP (Fig. 7d) and PtC (Fig. 7e), respectively. Both $m/z = 2$ ($H_2$) and $m/z = 3$ (DH) signals were detected for NGA-COF@Pt-NP and PtC, while only $m/z = 2$ ($H_2$) signal was collected for NGA-COF@Pt-2H during HER. This unambiguously demonstrates the presence of the residue of HER intermediates inside the bulk Pt and Pt particles. In contrast, with no DH or $D_2$ detected through in-situ DEMS, NGA-COF@Pt-2H exhibited excellent invulnerability and reproducibility towards HER intermediates. This also provided a valid explanation for the high TOF and mass activity towards HER, as well as the desirable stability of NGA-COF@Pt-2H at the atomic level. Notably, $D_2$ was not detected in the HER products of all samples, indicating that D atoms, rather than $D_2$ molecules, adsorb to the Pt (111) plane, which is consistent with previous studies[59,60].

To verify that NGA-COF@Pt-2H has the potential for further practical applications, NGA-COF@Pt-2H was assembled into a two-electrode

EWS cell as cathode, along with a RuTiIr dimensionally stable anode (DSA)[40] under simulating working condition (current density = 100 mA cm$^{-2}$, time = 260 hours). Supplementary Fig. 41 demonstrates the satisfactory stability of NGA-COF@Pt-2H since the final voltage is only 60 mV higher than the initial one in 1 M KOH after 260 h of catalysis. Although the catalyst was less stable under acidic conditions than under alkaline conditions, the value of the voltage increase (0.26 V) is still in an acceptable range since acidic conditions are theoretically more destructive to electrocatalysts, especially those COF-derived ones.

In summary, this work fabricated a conductive agent-free and pyrolysis-free single-atom HER catalyst NGA-COF@Pt by a simple and portable electrochemical modification strategy. The highly dispersed Pt single atoms and their corresponding precise chemical environments were investigated experimentally and theoretically. Furthermore, the low overpotential (13 mV), well-placed mass activity (18165 A g$_{Pt}^{-1}$), and turnover frequency (18.4 s$^{-1}$) were rationalized by revealing realistic NGA-COF@Pt-2H configuration with the help of in-situ electrochemical measurements and density functional theory (DFT) calculations. The complementary metal-support interaction between Pt single atoms and NGA-COF via Pt-N$_2$H$_2$ configuration greatly improves the conductivity and intrinsic activity of every Pt site without extra conductive agent or pyrolysis. Meanwhile, it maximizes the exposure of active sites and enhances the stability in acid by homogeneously combining the unique 2D COF nanosheets with Pt single atoms through a bottom-up strategy. These findings provide new insights for the design and synthesis of COF-derived SACs with high efficiency, sustainability, and cost-effectiveness.

## Methods

### Materials Synthesis

NGA-COF was prepared by a solvothermal method[61]. First, C3-symmetric 2,3,6,7,10,11-Triphenylenehexamine 6 HCl (TPHA, 21.48 mg, 0.04 mmol) and C3-symmetric Hexaketocyclohexane octahydrate (HKH, 12.50 mg, 0.04 mmol) were mixed by grinding them in a mortar. Then the mixed precursors, 0.9 mL of 1,3,5-Trimethylbenzene, 0.5 mL of 4 M acetic acid aqueous solution (oxygen was removed with argon) and 0.6 mL of 1,4-Dioxane were charged in a 20 mL glass tube (The inner diameter of the tube is 7.8 mm and the outer diameter is 10 mm). The tube was flashed frozen at 77 K (liquid nitrogen bath), degassed using three freeze-pump-thaw cycles and flame-sealed. The length of the pipe after flame-seal is 15 cm. The tube was heated at 140 °C for 3 days and the resulted dark brown precipitated product (NGA-COF) was further Soxhlet extracted in acetone to totally remove the impurities from the pores.

A sheet of titanium with TiO$_2$ nanotubes on surface (TiO$_2$ NTs) was fabricated by using the anodic oxidation method[62]. Initially, a sheet of titanium (20 mm × 10 mm × 0.1 mm) underwent ultrasonic cleaning in sequential baths of 10 mL acetone, 10 mL ethanol, and deionized water, each for 5 minutes. Subsequently, the titanium sheet was immersed in a mixed solution of HF:HNO$_3$:H$_2$O in a volume ratio of 1:4:5 for 15 seconds to remove the oxide layer. The prepared titanium sheet was then clamped to the electrode holder, serving as the anode, while a platinum electrode (1.5 × 1.5 cm$^2$) served as the cathode. Notably, no reference electrode was introduced during the anodization process. Both electrodes were placed in a 250 mL electrolytic cell, configured as a conventional beaker-type setup without an exchange membrane separating the cathode and anode. In the following text, unless otherwise specified, the type of electrolytic cell used is this beaker-type electrolytic cell without exchange membrane. The electrolyte consisted of a mixture of 10 ml NH$_4$F (0.33 g) aqueous solution and 90 mL glycol. Subsequent to immersing both electrodes in the electrolyte, the anodic oxidation process commenced, utilizing a constant voltage power supply set to 60 V for a duration of 120 min. After anodization, the sample was washed with deionized water and dried. Finally, titanium with TiO$_2$ nanotubes on surface was obtained and designated as TiO$_2$ NTs.

NGA-COF@Pt was prepared by an electrochemical modification strategy. Electrochemical modifications and measurements were performed by a CHI 660E electrochemical workstation with a three-electrode system, including a working electrode (WE), a counter electrode (CE), and a reference electrode (RE). When the electrolyte was 0.5 M H$_2$SO$_4$ solution, the Hg/Hg$_2$SO$_4$ (saturated K$_2$SO$_4$ electrolyte) electrode served as reference electrode (RE) while Hg/HgO (1 M KOH electrolyte) electrode was used in 1 M KOH solution. The experimental potential values were calibrated by using the following equation: $E$ vs. RHE = $E$ vs. Hg/Hg2SO4 + 0.64 + 0.059 pH; $E$ vs. RHE = $E$ vs. Hg/HgO + 0.098 + 0.059 pH. All electrode potentials were relative to reversible hydrogen electrodes (RHE) unless otherwise stated. We employed the automatic iR compensation function provided by the CHI 660E, which automatically compensates for the iR drop generated during testing. The compensation level was set to 100 %. Manual conductivity tests on the electrolyte were not conducted so as to avoid any interference with the response signal during electrochemical performance measurements.

5 mg of NGA-COF was dispersed in 900 μL of ethyl alcohol and 100 μL of 5 wt. % nafion mixed solution and ultrasonic for 30 min. 60 μL of liquid was pipetted onto a 1 × 1 cm$^2$ portion of the surface of carbon paper (CP, 1 × 2 cm$^2$) with natural drying (NGA-COF loading-0.6 mg cm$^{-2}$). Then a sheet of prepared CP and TiO$_2$ NTs (1 × 2 cm$^2$) was clamped to the platinum electrode holder and served as WE and CE, respectively. The electrolyte was 60 mL of 0.5 M H$_2$SO$_4$ containing a certain concentration of chloroplatinic acid (the mass of platinum obtained by ICP is about 3.988 ug). The volume of the electrolytic cell is 100 mL.

Electrochemical modification was carried out by using cyclic voltammetry (CV) conducted from −0.36 to −0.76 V with a scan rate of 20 mV s$^{-1}$ for 2000 segments with automatic iR compensation. Then NGA-COF@Pt was obtained on the CP substrate. After cyclic voltammetry, linear sweep voltammetry (LSV) was conducted more than 10 times from 0 to −0.2 V with a scan rate of 10 mV s$^{-1}$ with automatic iR compensation till the last two LSV curves were perfectly coincident so that WE surface reached a stable state. Then the WE was washed with deionized (DI) water thoroughly and dried naturally for further electrochemical measurements. NGA-COF@Pt-NP was obtained under the same conditions as above for comparison except for changing CV from 2000 to 2500 segments (Under this condition, Pt dispersed on COF in forms of nanoparticles rather than single atoms). NGA-COF@Pt-5000 was obtained under the same conditions as above except for changing CV from 2000 to 5000 segments. Besides, 5 mg of commercial 20 wt.% platinum on carbon (PtC, Tanaka Kikinzoku International KK, Japan) was dispersed in 900 μL of ethyl alcohol and 100 μL of 5 wt. % nafion mixed solution and ultrasonicated for 30 min to form a uniform ink. Then 60 μL of PtC ink was pipetted onto CP with natural drying for comparison (PtC loading-0.6 mg cm$^{-2}$).

### Physical characterizations

Scanning electron microscopy (SEM) was performed on a JEOL JSM-7800F with an operating voltage of 3 kV. Transmission electron microscopy (TEM) was performed on a Philips FEI Tecnai G2S-Twin microscope equipped with a field emission gun operating at 200 kV. X-ray photoelectron spectra (XPS) were recorded on a Thermo Fisher Scientific ESCALAB 250Xi unit with Al-Kα (1486.6 eV) as an X-ray source. The Raman spectroscopy was tested using a Renishaw Raman microscope with Ar-ion laser excitation ($\lambda$ = 514.5 nm). In-situ Raman analysis was conducted at 0.5 M H$_2$SO$_4$ or 1 M KCl solution. Each spectrum was accumulated with 10 s exposure time and 5 % of laser intensity. The Ag/AgCl (1 M KCl) and Pt wire were employed as the reference and counter electrode, respectively. The NGA-

COF@Pt electrode was used directly for in situ Raman spectro-electrochemical study. A LSV method was engaged at a scan rate of 0.3 mV s$^{-1}$. Pt L3-edge of X-ray absorption fine structure (XAFS) analyzes were performed with Si (111) crystal monochromators at the BL11B beamlines at the Shanghai Synchrotron Radiation Facility (SSRF) (Shanghai, China). The XAFS spectra were recorded at room temperature using a 4-channel Silicon Drift Detector (SDD) Bruker 5040. Pt L3-edge extended X-ray absorption fine structure (EXAFS) spectra were recorded in transmission mode. Negligible changes in the line-shape and peak position of Pt L3-edge XANES spectra were observed between two scans taken for a specific sample. The XAFS spectra of these standard samples (NGA-COF@Pt, Pt-foil, and PtO$_2$) were recorded in transmission mode. The spectra were processed and analyzed by the software codes Athena and Artemis[44]. Solid-state $^{13}$C NMR spectra were recorded on a Bruker AVANCE III spectrometer operating at 400 MHz for 1 h. The IR spectra were recorded using KBr pellets on a Bruker IFS-66 V/S FTIR spectrometer. X-ray diffraction (XRD) patterns were carried out using a Brucker D8 X-ray diffractometer with Cu Kα radiation (λ = 1.5418) at room temperature and accelerating voltage, and applied current were 40 kV and 20 mA, respectively. The diffraction data were recorded in the 2θ range of 3–80° with a scan rate of 5° per min. XRD data were processed by JADA 6 with the International Centre for Diffraction Data (ICDD-PDF) as database. Zeta potential was determined by Anton Paar surpass3. The solution for zeta potential measurement was 0.1 mmol/L KCl and the pH value was adjusted with 0.1 M HCl and 0.1 M NaOH. Gas adsorption−desorption analyzes were conducted using N$_2$ as adsorbent on an ASAP 2020 (Micromeritics instrument, United States) by nitrogen adsorption at 77 K. Inductively coupled plasma (ICP) data were collected from Agilent 725 (Agilent Technologies Inc).

### Electrochemical performance measurements

The as-prepared CP was employed as WE while the graphite rod (diameter is 6 mm) was applied as CE. Unless otherwise specified, WE and CE in the following text refer to these two electrodes. During the electrochemical testing process, the temperature was maintained at room temperature (20 to 30 °C), without any specific temperature control treatments. The volume of the electrolytic cell is 100 mL. The current density was normalized to the geometrical area (1 × 1 cm$^2$). The Hg/Hg$_2$SO$_4$ or Hg/HgO electrode served as RE in 0.5 M H$_2$SO$_4$ or 1 M KOH solution, as mentioned above. The volume of the electrolyte is 60 mL. Before electrochemical measurements, the reference electrodes were all calibrated by CV tests using glassy carbon electrode as standard WE[63]. Polarization curves were collected by conducting linear sweep voltammetry (LSV) tests at a scan rate of 10 mV s$^{-1}$. All LSV curves were measured with iR compensation at 100%. Tafel plots were converted from the LSV data at low overpotential fitted to the Tafel equation (η = b log j + a, where η is overpotential, j is the current density, a is the intercept of the y-axis, and b is the Tafel slope). Electrochemical impedance spectra (EIS) measurements were performed at different overpotential with frequencies from 1 Hz to 100 kHz and an amplitude of 5 mV without automatic iR compensation. The electrochemically active surface area (ECSA) was estimated by CV with a scan rate varying from 20 to 100 mV s$^{-1}$ in 0.5 M H$_2$SO$_4$ or 1 M KOH without automatic iR compensation[64]. Stabilities of the samples were evaluated by chronopotentiometric without automatic iR compensation. And the LSV curves of NGA-COF@Pt and PtC before and after chronopotentiometric tests were also recorded at a scan rate of 10 mV s$^{-1}$ with automatic iR compensation.

### Turnover frequency (TOF) calculations

Turnover frequency (TOF) of the Pt-based catalysts was calculated as the literature reported[7]. The TOF per Pt site of the fabricated Pt-based catalysts in this work was calculated by using the formula:

$$TOF(H_2 s^{-1}) = \frac{\text{total hydrogen turnover}}{\text{active sites}} \quad (1)$$

Total hydrogen turnover (geometric area = 1 cm$^2$):

$$\left(|j|\frac{mA}{cm^2}\right)\left(\frac{1\,Cs^{-1}}{1000\,mA}\right)\left(\frac{1\,mol\,e^-}{96485\,C}\right)\left(\frac{1\,mol}{2\,mol\,e^-}\right)$$
$$\left(\frac{6.022\times10^{23}\,H_2\,\text{molecules}}{1\,mol\,H_2}\right) \quad (2)$$

The number of catalytic active sites in NGA-COF@Pt was calculated according to the mass loading on the electrode (geometric area = 1 cm$^2$):

$$\left(\frac{\text{catalyst loading}\left(\frac{g}{cm^2}\right)\times Pt\,wt.\%}{Pt\,M_w\left(\frac{g}{mol}\right)}\right)\left(\frac{6.022\times10^{23}\,Pt\,\text{atoms}}{1\,mol\,Pt}\right) \quad (3)$$

### In-situ differential electrochemical mass spectrometric measurements

In-situ DEMS experiments were performed on an in-situ differential electrochemical mass spectrometer provided by Linglu Instruments (Shanghai) Co. Ltd. A typical test was carried out in a three-electrode cell with N$_2$ saturated 0.5 M H$_2$SO$_4$ as electrolyte. Firstly, the pristine samples were labeled with D isotopes in 60 mL of 0.5 M D$_2$SO$_4$ solution in D$_2$O by chronopotentiometric test for HER at 100 mA cm$^{-2}$ for 60 min without automatic iR compensation. The electrodes, loadings of catalysts and electrolytic cell of this chronopotentiometric test is identical to the electrochemical performance measurements mentioned above without automatic iR compensation. The resultant electrodes were then rinsed with H$_2$O for several times and dried in an oven overnight to remove the residual D$_2$O. The sample was placed into the sample chamber of the DMES instrument, and an injector was used to fill the sample chamber with electrolyte (0.5 M H$_2$SO$_4$ in H$_2$O). The total amount of electrolyte in the sample chamber, injection needle and outlet needle was about 10 mL. The air-tightness of the sample chamber was inspected and verified. The injection and outlet needles were meticulously adjusted to eliminate any air bubbles from the sample chamber, and the hermeticity of the sample chamber seal was checked and confirmed. After these processes, the in-situ DEMS measurement was carried out with applied potential on the samples. Chronopotentiometric test (2 mA cm$^{-2}$) was performed on D-labeled samples without automatic iR compensation for a specific period of time. The mass-to-charge ratio signal of the gas generated during this process was collected.

### EWS performances measurements

EWS cell was assembled by employing a classical two-electrode (WE and CE) system. The as-prepared CP loaded with NGA-COF@Pt (working area ~1 × 1 cm$^2$, loading~0.6 mg cm$^{-2}$) and ruthenium titanium iridium (RuTiIr) dimensionally stable anode (DSA working area ~3 × 3 cm$^2$) were used as cathode and anode, respectively[40]. The volume of the electrolytic cell is 250 mL and the volume of electrolyte is 200 mL. Stability tests of NGA-COF@Pt||RuTiIr electrode pair towards EWS in 0.5 M H$_2$SO$_4$ or 1 M KOH under simulating working condition were obtained by using constant current (current density = 0.1 A cm$^{-2}$) charging modes of Neware CT-4008T-5V6A-S1 without iR compensation.

## Faradaic efficiency (FE) calculations

FE was estimated by comparing the experimentally measured $H_2$ gas volume and the theoretical one. A chronopotentiometric test was carried out at $250\,mA\,cm^{-2}$ in 1 M KOH with modified CP as WE without automatic iR compensation. The WE and the CE (graphite rod) were isolated with anion exchange membrane (Nafion 211) in two different chambers of a H-type electrolytic cell. The volume of each chamber is 100 mL and the volume of electrolyte is 80 mL. Theoretical FE was calculated based on the following equation: FE = $(2 \times F \times n)/(i \times t)$, where F is the Faraday constant ($96485\,C\,mol^{-1}$); n is the numbers of moles of the produced $H_2$ (mol), respectively; i stands for the current (A); and t represents the electrolysis time (s).

## DFT calculations

Density Functional Theory(DFT): All ab initio total-energy calculations were carried out based on density functional theory (DFT) within the framework of VASP (Vienna Ab initio Simulation Packages) code, which uses a plane wave basis set for the electronic orbitals[65]. The electronic exchange and correlation were described within the generalized gradient approximation using the Perdew-Burke-Ernzerhof (PBE) functional[66]. The interaction of the valence electrons with the ionic cores was treated within the projector augmented-wave (PAW) method[67]. A Monkhorst-Pack grid was used by K-points $3 \times 3 \times 1$[68]. The cut-off energy was employed as 450 eV. All DFT calculations are to minimize all the residual forces until the convergence criterion of 0.02 eV/Å. A 20 Å vacuum layer is added perpendicular to the 2D system to eliminate the interaction between adjacent sheets. The empirical correction scheme of Grimme DFT-D3 was also performed[69], where the effect of vdW interactions was included explicitly.

The free energy diagrams for the $H_2$ evolution reaction were calculated, in which the chemical potential of a proton/electron ($H^+ + e^-$) is equal to half of that of one $H_2$ gas molecule. The change in free energy ($\Delta G_{H^*}$) in the overall transformation can be calculated as:

$$\Delta G_{H^{\cdot}} = E(\text{Surface} + H^*) - E(\text{Surface}) - 1/2 E(H_2) + \Delta E(\text{ZPE}) - T\Delta S$$

Where $E$ (surface + H*), $E$ (surface), $E(H_2)$, $\Delta E$ (ZPE), and $\Delta S$ represent the total energy for adsorption H, the energy of the bare surface, the energy of gaseous $H_2$, the zero-point energy change, and the entropy change between $H^+$ and $H_2$, respectively.

## Data availability

All data generated in this study are provided in the article and Supplementary Information, and the raw data generated in this study are provided in the Source Data file. Source data are provided with this paper.

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

## Acknowledgements

This work was financially supported by the National Natural Science Foundation of China (22271114), the Foundation of Science and Technology Development of Jilin Province, China (20200801004GH) and 111 Project (B17020). All fundings are awarded to Z. S. The authors also gratefully acknowledge the financial support by the program for JLU Science and Technology Innovative Research Team (JLUSTIRT). The authors would like to thank Yingluo Zhao from the University of Tokyo for the proofreading.

## Author contributions

Zi.Zh. conceived the idea, performed the experimental studies, and wrote the manuscript. Zh.Zh. conducted the theoretical calculations. C.C., M.X., and R.Z. performed the morphology characterization and analysis. R.W., S,W., and L.C. assisted in the synthesis. H.L., Y.H., W.X., Z.S., and S.F. supervised and directed the project.

## Competing interests

The authors declare no competing interests.
