## [Peer Review File · Nature Communications]

REVIEWER COMMENTS

Reviewer #1 (Remarks to the Author):

In this work, the authors developed a conductive agent-free and pyrolysis-free method to synthesize Pt single atoms anchored on nitrogen-rich graphene analogue COF, which regulating the HER activity and stability in full pH. Systemic characterizations and DFT calculations solidly demonstrated that the electronic structure could be modified by unique Pt-N₂ coordination environment. The results are interesting. Although some issues should be addressed in the revision process.

1. What unique structure of COF compare to other 2D materials such as graphen, layered double hydroxides (LDHs) and MOFs, supporting the corrosion resistance in acidic environments?
2. How do you prove that synthesized NGA-COF is AA stacking structure in line 143? Is there any more intuitive evidence?
3. How did you conduct the ICP experiment? Whether the Pt is completely dissolved is vital for the identifying of single atom density, which will determine the catalytic performance.
4. The coordination environment of Pt-N is very similar to that of Pt-O in XAS and is it possible to coordinate Pt-O₂ (not Pt-N₂) in COF? How did you rule out the Pt-O₂ coordination environment?
5. In line 207 and 208, it should be Fig. 3d and 3e not Fig. s3.
6. In general knowledge, the unsaturated coordination is unstable. In this work, the unsaturated Pt-N₂ is readily protonated to form Pt-N₂-H₂ structure in acidic media. How about in alkaline medium? Could you give some proof to explain it?
7. The unsaturated Pt-N₂ transform to Pt-N₂-H₂ structure in practical reaction process. Does this contradict with the M-N₄ or M-OH claimed in the introduction?
8. In Fig.7, DFT calculation is vital theory means to supporting the catalysts poisoned by HER intermediates.

Reviewer #2 (Remarks to the Author):

The authors presents a comprehensive study on COF-supported Pt SAC (NGA-COF@Pt) for HER, demonstrating an innovative approach in stabilizing single atom in COFs. Another interesting aspect of this paper is its avoidance of conductive agents and pyrolysis, marking an advancement in the field.

Moreover, the authors have thoroughly characterized the material using a range of techniques, providing clear evidence of the unique Pt-N₂ coordination, which is instrumental in achieving high catalytic activity and stability. The impressive performance of NGA-COF@Pt in various pH environments, coupled with its low overpotential and high turnover frequency, sets a new benchmark in the field. Additionally, the manuscript is well-structured, with clear and concise explanations that make the complex concepts accessible to readers. Therefore, I would recommend its publication after addressing the following minor issues:

1. In comparison to many previously described Pt-N₄ configurations, the Pt-N₂ configuration is expected to exhibit less stability. Could you compare their binding energies with some discussions?
2. What are the long-term stability and durability of the NGA-COF@Pt catalyst under continuous operation? Were there any degradation mechanisms observed?
3. Could you estimate the average distance between adjacent Pt sites? As indicated in recent studies, such as *J. Phys. Chem. Lett.* 2023, 14 (42), 9392-9402, it's suggested that adjacent single-atom sites might influence each other. Do you have any insights or discussion points on this matter?

Reviewer #3 (Remarks to the Author):

In this manuscript, the authors synthesized the first non-thermodynamically stable Pt-N₂ coordination active site through a mild electrochemical modification strategy. The definite structure ensures further study of metal-support interaction and corresponding HER catalytic mechanism. Given its unique asymmetric coordination environment and single-atom characteristic, NGA-COF@Pt exhibits excellent performance and stability against poisonous H⁺ intermediates in acid environment. Overall, the manuscript is well written, and the work is of novelty and significance. A few minor revisions are suggested before the manuscript can be accepted for publication in *Nature Communications*.

1. The authors need to explain why they prepared TiO₂ NTs and used them as CE during the preparation of catalysts.
2. It would be good to add the synthesis conditions including solvent, catalyst, temperature, and reaction time of NGA-COF in Fig. 1a.
3. Fig. 2d is not aligned, and there is a clear white line in the middle, which needs to be adjusted.
4. According to the simulative and experimental XRD and BET analysis in this manuscript, regular pore structure should be observed for this AA-stacked NGA-COF. However, in Fig. 3a, the atomic-resolution aberration-corrected HAADF-STEM image of NGA-COF@Pt did not show micropores of ~1.2 nm size. The reason for this should be explained.

5. In Fig. 5d and Supplementary Figure 18d, NGA-COF@Pt exhibits much higher R_{ct} than PtC. However, NGA-COF@Pt exhibits better performance towards HER compared to PtC, including overpotential, Tafel slope, TOF and mass activity. Please explain the reason.

6. In the Materials Synthesis section, the synthesis of NGA-COF was conducted in a 20 mL glass tube. However, after flame-seal, the volume of the tube must change. Therefore, the authors should clearly measure and declare the diameter and length of the tube after flame-seal, since this will directly affect the pressure of the system during the reaction process.

Dear Editors and Reviewers:

Thank you for your letter and comments concerning our manuscript entitled “*Single-Atom Platinum Sites with Asymmetric Coordination Environment on Fully Conjugated Covalent Organic Framework for Efficient Electrocatalysis*” (Manuscript ID: *NCOMMS-23-59946*). We have revised the manuscript based on your suggestions and highlighted the revisions for your review. The point-to-point responses to the reviewers’ comments are listed below:

Reviewer #1:

In this work, the authors developed a conductive agent-free and pyrolysis-free method to synthesize Pt single atoms anchored on nitrogen-rich graphene analogue COF, which regulating the HER activity and stability in full pH. Systemic characterizations and DFT calculations solidly demonstrated that the electronic structure could be modified by unique Pt-N₂ coordination environment. The results are interesting. Although some issues should be addressed in the revision process.

Comment 1:

What unique structure of COF compare to other 2D materials such as graphen, layered double hydroxides (LDHs) and MOFs, supporting the corrosion resistance in acidic environments?

Response 1:

a. For LDH, it is a kind of hydroxide, which is alkaline. The layered structure of LDH is stabilized by hydroxyl between the layers. Structural collapse and dissolution will occur in acidic environment, hindering their further application as electrocatalysts in acidic environment.

b. For most MOFs, the degree of reversibility of coordination bonds between metal nodes and ligands is much higher than that of covalent bond between ligands in covalent organic frameworks, which endows MOF higher crystallinity, but also makes MOF more vulnerable to harsh pH environments compared to COF. During the electrocatalytic process, the applied voltage damages the dynamic coordination bonds,

eventually leading to the reconstruction of the overall structure (ACS Catal. 2022, 12, 16, 10276–10284; ACS Energy Lett. 2019, 4, 4, 987–994). As a result, MOFs are typically used as pre-catalysts rather than catalysts in electrocatalysis. Some chemical modifications, such as pyrolysis, are necessary to stabilize the dynamic metal-ligand bonds, enabling their application in harsh pH environments.

c. Graphene, as one of the most representative two-dimensional conductive network materials, exhibits stability and excellent electrical conductivity under acidic conditions, making it an ideal substrate for anchoring noble-metal atoms. However, enhancing intrinsic activity and achieving an ideal metal-support interaction for regulating the electron structure of Pt single atoms on graphene remains challenging. Moreover, due to the structural stability of graphene, achieving uniform dispersion of noble-metal single atoms often requires high temperature and pressure, increasing the uncertainty of the final catalyst structure and hindering subsequent mechanism analysis.

d. In this work, the ligands of NGA-COF are linked by covalent imine bonds with high bond energy. Previous studies have confirmed the stability of these bonds in acidic conditions (Nat. Catal. 2022, 5, 414-429, Small 2019, 15, 1903643). In-situ Raman was carried out to demonstrate that the structure of NGA-COF@Pt remained intact in the HER process (**Fig. 5e and Supplementary Fig. 35**). In addition, abundant N coordination sites among NGA-COF allow Pt single atoms to be introduced more mildly at room temperature and pressure. Therefore, NGA-COF is a more ideal substrate for noble-metal single atoms in acidic environment than graphene, LDHs, and MOF.

Comment 2:

How do you prove that synthesized NGA-COF is AA stacking structure in line 143? Is there any more intuitive evidence?

Response 2:

Fig. 1 c XRD patterns of NGA-COF synthesized and simulated.

a. We have added the simulated XRD pattern of AB-stacking in **Fig. 1c**. The experimental result indicates that the NGA-COF synthesized in this work lacks the diffraction peak of the (101) crystal plane, a characteristic feature of AB-stacking. This absence confirms the AA-stacking interlayer chemical structure, consistent with previous reports (ACS Energy Lett. 2019, 4, 9, 2251–2258; Chem. Commun. 2019, 55, 9491-9494). We have added the following explanations in the manuscript:

The synthesized NGA-COF has a regular 2D layered structure with an interlayer distance of 3.37 Å.

The characteristic (101) peak of simulated AB-stacking was absent in the experimental XRD pattern, confirming an AA-stacking structure of the synthesized NGA-COF.

Fig. 1d Nitrogen adsorption–desorption isotherms of NGA-COF with corresponding pore size distribution inset.

Supplementary Fig. 2. Simulated pore size of NGA-COF.

b. According to nitrogen adsorption–desorption isotherms (**Fig. 1d**), the pore size distribution of the synthesized NGA-COF is concentrated in 11.8 Å, which is consistent with the theoretical simulation of AA-stacking (**Supplementary Fig. 2**). The pore size will be much smaller than 11.8 Å and can be detected by nitrogen adsorption–desorption isotherms if it is AB-stacking. This observation (**Fig. 1d**), aligned with the theoretical simulation of AA-stacking (**Supplementary Fig. 2**), provides additional confirmation of the AA-stacking structure in the synthesized NGA-COF.

Comment 3:

How did you conduct the ICP experiment? Whether the Pt is completely dissolved is vital for the identifying of single atom density, which will determine the catalytic performance.

Response 3:

High-resolution XPS spectra of Pt 4f of working electrode (NGA-COF@Pt) before and after acid treatment.

NGA-COF@Pt was treated with 2 mL of concentrated nitric acid under ultrasound for 24 hours, leading to the complete destruction of NGA-COF and dissolution of Pt into the solution. Subsequently, 48 mL of deionized water was added, and a 10 mL sample was extracted for the ICP test.

The accuracy of Pt content determination will affect the subsequent electrochemical performance data including normalization of TOF and Mass activity. To ensure that Pt would be completely dissolved in concentrated nitric acid without remaining on the carbon paper, XPS was performed on the carbon paper which had previously loaded with NGA-COF@Pt after digestion by concentrated nitric acid.

XPS spectra analysis (see above) following acid treatment revealed the absence of any Pt residue signals, confirming the complete dissolution of Pt from the working electrode into the solution.

Comment 4:

The coordination environment of Pt-N is very similar to that of Pt-O in XAS and is it possible to coordinate Pt-O₂ (not Pt-N₂) in COF? How did you rule out the Pt-O₂ coordination environment?

Response 4:
Fig. 3c k^2 -weighted FT-EXAFS spectra of Pt-foil, NGA-COF@Pt and PtO₂.

Fig. 3e Corresponding EXAFS R space fitting curve of NGA-COF@Pt (Inset: Simulations models of NGA-COF@Pt).

Path	CN	R(Å)	ΔE_0 (eV)	$\sigma^2(10^{-3} \text{ \AA}^2)$	R factor
Pt-N	2	1.97±0.012	4.2±0.018	6.5±3.5	0.0147

Supplementary Table 1. Structural parameters obtained from the Pt L₃-edge EXAFS fitting for NGA-COF@Pt.

a. Given the close resemblance between the lengths of Pt-N and Pt-O bonds, the R values in the k²-weighted FT-EXAFS analysis (**Fig. 3c**) are remarkably similar. Nonetheless, a subtle difference is discernible, with the R value of Pt-N being slightly less than that of Pt-O, as clearly shown in **Fig. 3c**.

This is consistent with previous study (please refer to Fig. 2h and Table S2 of Angew. Chem. Int. Ed. 2021, 60, 19262). In addition, the error between experimental and simulated result is small enough (R factor = 1.47 % < 2 %) in the R-space fitting of EXAFS (**Fig. 3e and Supplementary Table 1**), indicating that the coordination environment of Pt single-atom in synthesized NGA-COF@Pt is consistent with the simulated Pt-N₂ model provided inset **Fig. 3e** within the allowable error range. Therefore, although the coordination environment of Pt-N is very similar to that of Pt-O, it can still be confirmed that the coordination environment of Pt in NGA-COF@Pt is Pt-N₂ rather than Pt-O₂ by XAS fine spectrum analysis.

Supplementary Fig. 11. (b) High-resolution XPS spectra of O 1s of NGA-COF and NGA-COF@Pt.

Fig. 4e High-resolution XPS spectra of N 1s of different samples.

Fig. 4a ELFs of NGA-COF and *b* NGA-COF@Pt with Bader charge analysis marked on specific atoms. The H, C, N, and Pt elements are shown in green, grey, blue and red, respectively. *c* The differential charge density distribution map of NGA-COF@Pt along the *c* (upper), *a* (bottom left) and *b* (bottom right) axis, where the isosurface value is set to be $0.005 e \text{ \AA}^{-3}$ and the positive and negative charges are shown in yellow and cyan, respectively.

b. The high-resolution XPS spectra of O 1s of NGA-COF and NGA-COF@Pt was added in **Supplementary Fig. 11b**, indicating that the chemical environments of O remained the same after the introduction of Pt. This confirms that Pt was not directly coordinated with O, ruling out the Pt-O₂ coordination environment. In addition, after the introduction of Pt, there was a notable change in the binding energy of N (**Fig. 4e**). Also, the observed trend in the binding energy shift aligns with the theoretical

calculations, as illustrated in **Fig. 4a-c**. All these confirm Pt-N₂ rather than Pt-O₂ configuration.

Comment 5:

In line 207 and 208, it should be Fig. 3d and 3e not Fig. s3.

Response 5:

Thanks for your careful observation. Now the figure reference has been corrected (Fig. 3d and 3e and Supplementary Table 1). A similar error was identified in line 371 (Supplementary Fig. S37), and has been corrected as well (Supplementary Fig. 37).

Comment 6:

In general knowledge, the unsaturated coordination is unstable. In this work, the unsaturated Pt-N2 is readily protonated to form Pt-N2-H2 structure in acidic media. How about in alkaline medium? Could you give some proof to explain it?

Response 6:

In this work, Pt-H formation took place after electrochemical modification and before the HER test. Specifically, NGA-COF@Pt-2H was directly synthesized through electrochemical modification of NGA-COF in acidic media in a single step, as opposed to a two-step process involving initial Pt anchoring among NGA-COF and subsequent protonation of Pt-N2 with 2H. This conclusion has been explained in detail in the manuscript (from line 340 to line 362, **Fig. 6d, Supplementary Figs. 32-36**).

Therefore, in alkaline medium, the catalytic site of HER is Pt-N2-H2 rather than Pt-N2. The subsequent discussion presents both experimental and theoretical evidence to verify that Pt-N2-H2 will not undergo further protonation in an alkaline medium.

Fig. 6d Zeta potential of NGA-COF and NGA-COF@Pt at different pH values.

Supplementary Fig. 27. Configurations of H* intermediate adsorbed on the Pt site of NGA-COF@Pt, NGA-COF@Pt-H and NGA-COF@Pt-2H, respectively.

a. From an experimental point of view, Zeta potential measurements conducted at different pH values (**Fig. 6d**) confirm that NGA-COF is readily protonated at low pH condition given its high zeta potential. However, for NGA-COF@Pt, the Zeta potential is around 0 in acidic, neutral and alkaline environments, indicating that negligible charge interaction was detected at the interface between the working electrode (NGA-COF@Pt) and the electrolyte. This phenomenon directly rules out the possibility of protonation under alkaline conditions.

b. From a theoretical calculation standpoint, the adsorption energy of Pt site of Pt-N₂ towards the first two hydrogen atoms is particularly negative (-1.217 eV and -1.254 eV). The formation of Pt-N₂-H₂ is not reversible, which makes it difficult to go back to Pt-N₂. Therefore, NGA-COF@Pt will not undergo protonation under alkaline conditions.

Comment 7:

The unsaturated Pt-N₂ transform to Pt-N₂-H₂ structure in practical reaction process. Does this contradict with the M-N₄ or M-OH claimed in the introduction?

Response 7:

Fig. 7b (old version) Schematic illustration of H intermediates adsorbed on the Pt site of NGA-COF@Pt-2H, which is easy to desorb. DEMS measurements of H₂, DH and D₂ signals from the reaction products for D-labeled.

The Pt-N₂-H₂ structure is neither competitive nor contradictory to M-N₄ and M-OH. Instead, this asymmetric coordination configuration of the Pt single-atom serves as a novel complement to the existing configurations of electrocatalytic sites.

NGA-COF@Pt-2H, of which Pt forms an asymmetric square planar geometry with two coordinated N and H atoms, not only prevents the generation of Pt (111) crystal surface and its corresponding defects, but also avoids the over-strong specific adsorption of intermediate H through the H-saturated Pt tetra coordination crystal field (D_{4h}). In other words, H atoms that may be adsorbed at the Pt sites of NGA-COF@Pt-2H during HER will be more easily detached because Pt tends to recover from unstable Pt-N₂H₃ to Pt-N₂H₂ tetrahedral configuration (**Fig. 7b**).

Comment 8:

In Fig.7, DFT calculation is vital theory means to supporting the catalysts poisoned by HER intermediates.

Response 8:

Fig. 7 Schematic illustration and corresponding free energy diagram of HER over Pt site of **a** Pt (111) and **b** NGA-COF@Pt before and after optimizing by H* intermediates.

Thanks for your suggestion. The DFT calculations on adsorption free energy of HER intermediates over different Pt sites before and after hydrogenation have been added to **Fig. 7a-b**. For Pt (111), the binding energy of the Pt site toward H* decreases from -0.224 eV to -0.237 eV after hydrogenation. For NGA-COF@Pt, the binding energy of Pt site towards H* increases from -1.217 eV to -0.019 eV after endowing Pt-N₂ configuration with two H atoms. In conclusion, theoretical evidence demonstrates that hydrogenation poisons the traditional Pt (111) site, while optimizing the Pt-N₂ site of the NGA-COF@Pt electrocatalyst through DFT calculations.

Reviewer #2:

The authors presents a comprehensive study on COF-supported Pt SAC (NGA-COF@Pt) for HER, demonstrating an innovative approach in stabilizing single atom in COFs. Another interesting aspect of this paper is its avoidance of conductive agents and pyrolysis, marking an advancement in the field. Moreover, the authors have thoroughly characterized the material using a range of techniques, providing clear evidence of the unique Pt-N₂ coordination, which is instrumental in achieving high catalytic activity and stability. The impressive performance of NGA-COF@Pt in various pH environments, coupled with its low overpotential and high turnover frequency, sets a new benchmark in the field. Additionally, the manuscript is well-structured, with clear and concise explanations that make the complex concepts accessible to readers. Therefore, I would recommend its publication after addressing the following minor issues:

Comment 1:

In comparison to many previously described Pt-N₄ configurations, the Pt-N₂ configuration is expected to exhibit less stability. Could you compare their binding energies with some discussions?

Response 1:

Supplementary Fig. 9. Formation paths and corresponding calculated formation energies of different samples.

The formation path and corresponding theoretical calculations, as required, are provided in **Supplementary Fig. 9**. A detailed discussion of this process is included in the **Supplementary Note for Fig. 9**. In short, the acidic environment and reduction voltage during the electrochemical modification promote the transition of the unstable NGA-COF@Pt (Pt-N₂) intermediate to the more stable NGA-COF@Pt-2H (Pt-N₂H₂) configuration.

In addition, it is worth noting that the formation of Pt-H occurred after the electrochemical modification and before the HER test. In other words, NGA-COF@Pt-2H was directly synthesized by electrochemical modifying NGA-COF in acidic media in one step rather than anchoring Pt among NGA-COF firstly and protonating Pt-N₂ with 2H secondly. This conclusion has been explained in detail in the manuscript (from line 340 to line 362, **Fig. 6d, Supplementary Figs. 32-36**).

Comment 2:

What are the long-term stability and durability of the NGA-COF@Pt catalyst under continuous operation? Were there any degradation mechanisms observed?

Response 2:

Supplementary Fig. 15. a Chronopotentiometric tests and b corresponding polarization curves of different samples before and after chronopotentiometric tests in 0.5 M H₂SO₄.

Supplementary Fig. 22. a Chronopotentiometric tests and b corresponding polarization curves of different samples prior to and after chronopotentiometric tests in 1 M KOH.

Supplementary Fig. 41. Stability tests of NGA-COF@Pt-2H||RuTiIr electrode pair towards EWS in 0.5 M H₂SO₄ or 1 M KOH under simulating working condition (constant current charging, current density = 100 mA cm²).

a. The long-term stability and durability of the NGA-COF@Pt catalyst have been evaluated by chronopotentiometric and linear sweep voltammetry tests in acidic (Supplementary Fig. 15) and alkaline (Supplementary Fig. 22) environments. In addition, a continuous operation stability test under simulating working condition (current density = 100 mA cm⁻², time = 260 hours) was carried out in a two-electrode EWS cell where the NGA-COF@Pt catalyst was employed as the cathode (Supplementary Fig. 41). The stability of NGA-COF@Pt is demonstrated by a final voltage increase of only 60 mV after 260 hours of HER in 1 M KOH. Although the catalyst was less stable under acidic conditions than under alkaline conditions, the value of the voltage increase (0.26 V) is still in an acceptable range since acidic conditions are theoretically more destructive to electrocatalysts, especially those COF-derived ones.

Supplementary Fig. 16. a-d SEM images and e-f TEM images at different magnifications of NGA-COF@Pt after chronopotentiometric test in 0.5 M H₂SO₄.

Supplementary Fig. 23. *a-d* SEM images and *e-f* TEM images at different magnifications of NGA-COF@Pt after chronopotentiometric tests in 1 M KOH.

b. The morphology of the NGA-COF@Pt catalyst has been observed by SEM and TEM after stability tests in acidic (**Supplementary Fig. 16**) and alkaline (**Supplementary Fig. 23**) environments. In SEM images, the size and arrangement of catalyst particles hardly changed after stability tests both in acidic and alkaline environments. In TEM images, the layered structure of COF was also retained. And Pt single atom sites did not agglomerate into Pt particles. Therefore, it can be concluded that the morphology and physical structure of the catalyst did not change significantly before and after the stability test.

Fig. 5e In-situ Raman spectra of NGA-COF@Pt at various potentials. Electrolyte: 0.5 M H₂SO₄.

Supplementary Fig. 35. In situ Raman analysis of NGA-COF@Pt during the HER process at different potentials in 1 M KCl.

c. The in-situ Raman was carried out to investigate the transformation of chemical bond of NGA-COF@Pt during the HER process (**Fig. 5e and Supplementary Fig. 35**). The Raman shift at around 1356 cm⁻¹, 1462 cm⁻¹, 1513 cm⁻¹, and 1610 cm⁻¹ corresponds to the characteristic peaks of the benzene ring, channel relaxation, C=C, and C=N vibrations, respectively. It can be concluded that even if the applied voltage of the working electrode (NGA-COF@Pt) reaches -1.8 V, its chemical structure remains intact since every characteristic peak is preserved under continuous operation.

In summary, no significant changes were observed in the physical structure and chemical environment of NGA-COF@Pt after stability test based on the above characterization methods. This is also why NGA-COF@Pt is more stable and endurable than commercial PtC. The negligible attenuation of its performance can be attributed to inevitable concentration polarization and electrochemical polarization.

Comment 3:

Could you estimate the average distance between adjacent Pt sites? As indicated in recent studies, such as J. Phys. Chem. Lett. 2023, 14 (42), 9392-9402, it's suggested that adjacent single-atom sites might influence each other. Do you have any insights or discussion points on this matter?

Response 3:

Simulated distance between adjacent Pt sites of NGA-COF@Pt.

Simulated distance between adjacent M sites of M2-N4 configuration.

Supplementary Fig. 6. *a* Magnified atomic-resolution HAADF-STEM image of NGA-COF@Pt. *b* Enlarged atomic-resolution HAADF-STEM image of NGA-COF@Pt and *c* corresponding distance between two bright spots (Pt atoms) in the red box. *d* Simulated distance between two adjacent Pt atoms in NGA-COF@Pt.

a. According to the optimization result of theoretical structure simulation, the distance between adjacent Pt single atoms is about 12.5 Å. In addition, magnified atomic-resolution HAADF-STEM image of NGA-COF@Pt demonstrates that the spacing between most Pt single atoms is about 1 to 2 nm (**Supplementary Fig. 6**), which is close to the simulation results. It is worth noting that the COF material, unlike traditional inorganic porous materials, cannot withstand intense electron beam focusing. The irreversible damage caused by the strong electron beam of atomic-resolution TEM may lead to the twist and deformation of COF, resulting in a slight discrepancy between experimental and simulated results.

b. We have read the references you provided carefully (J. Phys. Chem. Lett. 2023, 14 (42), 9392-9402, <https://doi.org/10.1021/acs.jpcclett.3c02273>) and understand that

adjacent single-atom sites might influence each other through the interaction and migration of electron clouds. However, in our work, the density of Pt single atoms is not large enough. The distance between adjacent Pt (12.5 Å) is greater than that between metals in the M₂-N₄ configuration (7.4 Å). As the distance increases, the intensity of the electron interaction between adjacent Pt atoms decreases exponentially. Therefore, in this work, the interaction of the electronic structure between adjacent Pt atoms is much less than that of the metal-support interaction between Pt and NGA-COF substrate.

c. There are instances of individual adjacent Pt atoms with close distances. For instance, in **Supplementary Fig. 6**, two adjacent Pt atoms at a distance of 5.8 Å have been observed. However, this thermodynamically unstable configuration is almost impossible to exist either theoretically or experimentally (see the additional aberration-corrected HAADF-STEM images of NGA-COF@Pt below) and cannot represent the overall structure of the NGA-COF@Pt catalyst.

To sum up, the more thermodynamically-stable NGA-COF@Pt (**Supplementary Table 2**) was finally chosen as theoretical modeling for DFT calculation to determine adsorption energy of H* intermediates and to investigate HER mechanism. This modeling comprehensively takes into account both the mathematical convergence of theoretical calculation and the realism of reaction process monitored by experiment.

Additional aberration-corrected HAADF-STEM images of NGA-COF@Pt.

Reviewer #3:

In this manuscript, the authors synthesized the first non-thermodynamically stable Pt-N₂ coordination active site through a mild electrochemical modification strategy. The definite structure ensures further study of metal-support interaction and corresponding HER catalytic mechanism. Given its unique asymmetric coordination environment and single-atom characteristic, NGA-COF@Pt exhibits excellent performance and stability against poisonous H⁺ intermediates in acid environment. Overall, the manuscript is well written, and the work is of novelty and significance. A few minor revisions are suggested before the manuscript can be accepted for publication in Nature Communications.

Comment 1:

The authors need to explain why they prepared TiO₂ NTs and used them as CE during the preparation of catalysts.

Response 1:

Our previous work (Chem. Sci. 2022, 13, 8876-8884) has proposed a mild electrochemical modification strategy by employing TiO₂ NT as CE. Specifically, TiO₂ NT CE creates a localized electric field that allows metal atoms, including Fe, Mo, Ni, Cu and Pt, to embed uniformly, precisely and robustly into the metal-organic framework (MOF)-derived carbon matrix. Previous studies have demonstrated that TiO₂ NTs provide a confined electric field, resulting in more even and regular arrangement of deposited particles on the working electrode. This electrochemical modification method, previously successful with metal-organic framework-derived carbon matrices, is extended to the COF substrate in the current study. The SEM images of TiO₂ NT CE used in this work are presented below:

Supplementary Fig. 3. a-c SEM images of TiO₂ NTs.

Comment 2:

It would be good to add the synthesis conditions including solvent, catalyst, temperature, and reaction time of NGA-COF in Fig. 1a.

Response 2:

Fig. 1a Schematic illustration of the synthesis of NGA-COF.

The synthesis conditions have been incorporated into **Fig. 1a**, including solvent, catalyst, temperature, and reaction time of NGA-COF.

Comment 3:

Fig. 2d is not aligned, and there is a clear white line in the middle, which needs to be adjusted.

Response 3:

Fig. 2d EDS mappings of NGA-COF@Pt and commercial 20 wt.% PtC for Pt elements.

The alignment issue in **Fig. 2d** has been fixed according to the suggestion.

Comment 4:

According to the simulative and experimental XRD and BET analysis in this manuscript, regular pore structure should be observed for this AA-stacked NGA-COF. However, in Fig. 3a, the atomic-resolution aberration-corrected HAADF-STEM image of NGA-COF@Pt did not show micropores of ~ 1.2 nm size. The reason for this should be explained.

Response 4:
Additional aberration-corrected HAADF-STEM images of NGA-COF@Pt.

Fig. 1d. Nitrogen adsorption–desorption isotherms of NGA-COF with corresponding pore size distribution inset.

Here are the rationales:

- a. Due to the numerous layers stacked in the c-axis direction of COF, a deviation of just 1° in the electron beam incidence angle can prevent the electron beam from passing through the COF and imaging on the receiver. This means that the electronic gun and the sample need to form a very precise angle, which is often difficult to achieve by manipulating the sample or filming process.
- b. Unlike traditional inorganic zeolite materials, the COF material itself is not able to withstand very intense electron beam focusing. The damage to COF by electron beam is irreversible, which is likely to cause the interlayer deviation of COF, resulting in the decrease of sample crystallinity.
- c. There are currently some COFs that are stable enough to be photographed with clear channel structure (J. Am. Chem. Soc. 2022, 144, 12400-12409). The purpose of these works is to observe COF crystals at the atomic level. However, the synthesized NGA-COF in this paper is intended to design a conductive agent-free and pyrolysis-free platinum single atom catalyst for HER under full pH condition. So how to maintain stability in the electron beam is beyond the scope of this article. Besides, after the coordination of Pt, the structure of COF has undergone great changes compared with the original structure (**Supplementary Fig. 7**). Therefore, it is almost impossible to photograph such atomic-scale pore structures under aberration-corrected HAADF-

STEM.

d. Since the HAADF-STEM imaging did not work as expected after many attempts, nitrogen adsorption-desorption isotherms of NGA-COF was carried out (**Fig. 1d**) to quantify the pore size distribution of the NGA-COF from a bulky viewpoint. It is evident that the pore size of NGA-COF is around 1.2 nm, which is in agreement with the simulations and previous literature. This approach, while unable to capture atomic-scale pore structures, provides a meaningful quantification of the pore size distribution in NGA-COF from a broader perspective.

Comment 5:

In Fig. 5d and Supplementary Fig. 18d, NGA-COF@Pt exhibits much higher R_{ct} than PtC. However, NGA-COF@Pt exhibits better performance towards HER compared to PtC, including overpotential, Tafel slop, TOF and mass activity. Please explain the reason.

Response 5:

Supplementary Fig. 37. Calculated TDOS of NGA-COF and NGA-COF@Pt.

Fig. 5d EIS of different samples. Electrolyte: 0.5 M H_2SO_4 .

Supplementary Fig. 18. d EIS of different samples in 1 M KOH.

a. First, the final performance of the catalyst is influenced by multiple factors, with electron transfer resistance being just one of them. In other words, a linear relationship between electron transfer resistance and performance does not exist.

b. Second, after the doping of Pt single atoms, the band gap was reduced and the connectivity of energy band was improved significantly (**Supplementary Fig. 37**). In addition, the overall electron connectivity was also improved experimentally (**Fig. 5d and Supplementary Fig. 18d**). However, there is still a gap in conductivity between the fully conjugated COF skeleton and the conductive graphite structure of PtC. Although the Pt single atom improves the electron transport capacity of the local Pt-N-C unit, limited Pt load (≈ 2.66 wt.%) did not activate the overall COF skeleton.

c. Thirdly, NGA-COF@Pt has better performance thanks to its higher intrinsic activity. Specifically, the TOF and Mass activity of each Pt site in NGA-COF@Pt is much higher than that in PtC (**Fig. 5b and Supplementary Fig. 18b**). In addition, lower Tafel slopes of NGA-COF@Pt compared to PtC in both acidic (**Fig. 5c and Supplementary Fig. 18b**) and alkaline environments suggest higher HER kinetics, which also leads to better performance.

Comment 6:

In the Materials Synthesis section, the synthesis of NGA-COF was conducted in a 20 mL glass tube. However, after flame-seal, the volume of the tube must change. Therefore, the authors should clearly measure and declare the diameter and length of the tube after flame-seal, since this will directly affect the pressure of the system during the reaction process.

Response 6:

The synthesis method has been modified according to the requirements. For details, please see the highlighted section below:

Then the mixed precursors, 0.9 mL of 1,3,5-Trimethylbenzene, 0.5 mL of 4 M acetic acid aqueous solution (oxygen was removed with argon) and 0.6 mL of 1,4-Dioxane were charged in a 20 mL glass tube (The inner diameter of the tube is 7.8 mm and the outer diameter is 10 mm). The tube was flashed frozen at 77 K (liquid nitrogen bath), degassed using three freeze-pump-thaw cycles and flame-sealed. The length of the pipe after flame-seal is 15 cm.

REVIEWERS' COMMENTS

Reviewer #1 (Remarks to the Author):

The revision is improved accordingly and suitable for acceptance, no more comments.

Reviewer #2 (Remarks to the Author):

The authors have satisfactorily addressed my concerns, and I am pleased to recommend their manuscript for publication.

Reviewer #3 (Remarks to the Author):

The authors have addressed all my concerns. I recommend publication of this manuscript in the journal.

Reviewer #1 (Remarks to the Author):

The revision is improved accordingly and suitable for acceptance, no more comments.

Reviewer #2 (Remarks to the Author):

The authors have satisfactorily addressed my concerns, and I am pleased to recommend their manuscript for publication.

Reviewer #3 (Remarks to the Author):

The authors have addressed all my concerns. I recommend publication of this manuscript in the journal.